# OXA1L mutations cause mitochondrial encephalopathy and a combined oxidative phosphorylation defect

Kyle Thompson[1], Nicole Mai[1], Monika Oláhová[1], Filippo Scialó[2], Luke E Formosa[3], David A Stroud[4], Madeleine Garrett[3], Nichola Z Lax[1], Fiona M Robertson[1], Cristina Jou[5], Andres Nascimento[6], Carlos Ortez[6], Cecilia Jimenez-Mallebrera[6], Steven A Hardy[1], Langping He[1], Garry K Brown[7], Paula Marttinen[8], Robert McFarland[1] [iD], Alberto Sanz[2], Brendan J Battersby[8] [iD], Penelope E Bonnen[9], Michael T Ryan[3], Zofia MA Chrzanowska-Lightowlers[1], Robert N Lightowlers[1] & Robert W Taylor[1,*] [iD]

## Abstract

OXA1, the mitochondrial member of the YidC/Alb3/Oxa1 membrane protein insertase family, is required for the assembly of oxidative phosphorylation complexes IV and V in yeast. However, depletion of human OXA1 (OXA1L) was previously reported to impair assembly of complexes I and V only. We report a patient presenting with severe encephalopathy, hypotonia and developmental delay who died at 5 years showing complex IV deficiency in skeletal muscle. Whole exome sequencing identified biallelic OXA1L variants (c.500_507dup, p.(Ser170Glnfs*18) and c.620G>T, p.(Cys207Phe)) that segregated with disease. Patient muscle and fibroblasts showed decreased OXA1L and subunits of complexes IV and V. Crucially, expression of wild-type human OXA1L in patient fibroblasts rescued the complex IV and V defects. Targeted depletion of OXA1L in human cells or Drosophila melanogaster caused defects in the assembly of complexes I, IV and V, consistent with patient data. Immunoprecipitation of OXA1L revealed the enrichment of mtDNA-encoded subunits of complexes I, IV and V. Our data verify the pathogenicity of these OXA1L variants and demonstrate that OXA1L is required for the assembly of multiple respiratory chain complexes.

**Keywords** encephalopathy; insertase; mitochondria; OXA1L; OXPHOS
**Subject Categories** Genetics, Gene Therapy & Genetic Disease; Neuroscience

## Introduction

Mitochondrial disorders encompass a wide range of clinical phenotypes and are genetically heterogeneous. Mitochondrial dysfunction can arise from mutations in either the maternally inherited mitochondrial genome (mtDNA) or in nuclear genes that encode mitochondrial proteins. The 13 mtDNA-encoded polypeptides are essential components of the oxidative phosphorylation (OXPHOS) system. Many nuclear genetic defects causing mitochondrial disease are in gene products required for maintaining correct expression of mtDNA. These include proteins involved in replication, mtDNA maintenance, transcription, mtRNA processing, maturation and translation (Lightowlers et al, 2015).

In recent years, next-generation sequencing (NGS) technologies, particularly whole exome sequencing (WES), have proven successful in identifying an increasing number of mutations in genes causing primary mitochondrial disorders (Calvo et al, 2012; Taylor et al, 2014; Wortmann et al, 2015; Kohda et al, 2016; Pronicka et al, 2016). There are currently more than 250 nuclear-encoded genes associated with mitochondrial disease (Mayr et al, 2015; Alston et al, 2017; Frazier et al, 2017). However, considering the mitochondrial proteome is estimated to include approximately 1,200 proteins, it is likely that there are many more candidate genes still to be described. One such gene is OXA1L, which has long been proposed as a candidate gene to be screened for mutations in patients presenting with combined respiratory chain deficiencies (Rotig et al, 1997). To date, no mutations have been linked to mitochondrial disease (Coenen et al, 2005).

1 Wellcome Centre for Mitochondrial Research, Newcastle University, Newcastle upon Tyne, UK
2 Institute for Cell and Molecular Biosciences, Newcastle University Institute for Ageing, Newcastle University, Newcastle upon Tyne, UK
3 Department of Biochemistry and Molecular Biology, Monash Biomedicine Discovery Institute, Monash University, Melbourne, Vic., Australia
4 Department of Biochemistry and Molecular Biology and The Bio21 Molecular Science and Biotechnology Institute, The University of Melbourne, Parkville, Vic., Australia
5 Pathology Department, Hospital Sant Joan de Déu, CIBERER, Barcelona, Spain
6 Neuromuscular Unit, Neuropaediatrics Department, Hospital Sant Joan de Déu, CIBERER - ISCIII, Barcelona, Spain
7 Oxford Medical Genetics Laboratories, Churchill Hospital, Oxford University Hospitals NHS Foundation Trust, Oxford, UK
8 Institute of Biotechnology, University of Helsinki, Helsinki, Finland
9 Department of Molecular and Human Genetics, Baylor College of Medicine, Houston, TX, USA
*Corresponding author. Tel: +44 191 2083685; E-mail: robert.taylor@ncl.ac.uk

OXA1L is a member of the YidC/Alb3/Oxa1 membrane protein insertase family (Hennon *et al*, 2015). The yeast orthologue of OXA1L, Oxa1p, was identified as an important factor for the assembly of complex IV (Bonnefoy *et al*, 1994a) and has also been shown to be important for the assembly of complex V (Altamura *et al*, 1996). Oxa1p interacts with the mitoribosome (Jia *et al*, 2003; Kohler *et al*, 2009) and is required for the co-translational membrane insertion of mitochondrial-encoded Atp6p, Atp9p, Cox1p, Cox2p, Cox3p and Cytb (Hell *et al*, 1997, 1998, 2001; Jia *et al*, 2007). Oxa1p also appears to have a direct role in the insertion of nuclear-encoded inner mitochondrial membrane (IMM) proteins (Hell *et al*, 2001) including Oxa1p itself (Hell *et al*, 1998) and an indirect effect on many more IMM proteins including several metabolite transporters (Hildenbeutel *et al*, 2012), since Oxa1p is crucial for the biogenesis of the Tim18-Sdh3 module of the carrier translocase (Stiller *et al*, 2016).

The majority of studies into the function of Oxa1 have been conducted in yeast. Human OXA1L shares 33% identity with the Oxa1p yeast protein and was identified as a homologue by functional complementation and expression of the human gene into an Oxa1p null strain of *Saccharomyces cerevisiae*. This partially rescued the phenotype of impaired cytochrome *c* oxidase (COX) assembly, suggesting that OXA1L likely performs a similar role in human cells (Bonnefoy *et al*, 1994b). Indeed, human OXA1L has since been reported to bind to the mammalian mitoribosome via its C-terminal tail (Haque *et al*, 2010). In contrast to yeast, shRNA-mediated knockdown of *OXA1L* in human cells was shown to cause a defect in complex I and complex V assembly, but did not affect complex IV (Stiburek *et al*, 2007). As other insertases may be present in mitochondria, clarification into the importance of OXA1L is required.

Here, we present the clinical, biochemical and molecular characterisation of a patient with a severe, childhood-onset mitochondrial encephalopathy and combined respiratory chain deficiency due to biallelic variants in *OXA1L* identified by WES. Results from cellular and biochemical approaches suggest that OXA1L plays a major role as the insertase for the biogenesis of respiratory chain complexes.

# Results

## Case report

The index case, a 5-year-old male, was born to non-consanguineous healthy Chinese parents. Three previous pregnancies had resulted in miscarriages in the first and second trimesters without obvious cause, but this pregnancy had been uneventful, though delivery was complicated by a clavicular fracture. He had a good birthweight of 4.1kg and did not require resuscitation with Apgars recorded as 9[1] and 10[5]. He showed signs of severe hypotonia from birth with subsequent neurodevelopmental delay, achieving independent sitting at 12 months, but never being able to stand or walk. Language skills were also severely delayed in that he was unable to understand even simple instructions and made no attempt to speak or supplement communication with non-verbal behaviour. He was reliant on parents for all activities of daily living. Obstructive sleep apnoea was confirmed by polysomnography at the age of 3 years, and he had a tonsillectomy prior to commencing non-invasive nocturnal ventilation. On examination at 4 years, he was noted to be obese (32 kg) and exhibited generalised weakness, hypotonia and areflexia in his lower limbs. Iron deficiency anaemia was identified though the cause was unclear. Brain MRI revealed dysgenesis of the corpus callosum but was otherwise normal (Fig EV1). Electrophysiological testing showed normal motor nerve velocities, but low amplitude CMAPs and a neurogenic pattern on electromyography. At 5 years, he presented with a brief febrile illness associated with a mild metabolic acidosis (venous lactate 2.48 mmol/l, normal range 0.7–2.1 mmol/l) and leucocytosis. Further metabolic workup revealed increased serum alanine (520 μmol/l; normal range < 416), but ammonia, CDG and biotinidase activity were normal, as was PDHc activity in patient fibroblasts. Acylcarnitines and urinary organic acids were not determined. His condition deteriorated rapidly with generalised seizures and encephalopathy prior to a cardiorespiratory arrest from which he could not be resuscitated. An older female sibling, also considered to have neurodevelopmental delay, died in China aged 12 months. This death was also preceded by a febrile illness, but the cause remains unclear. A younger male sibling was born following pre-natal testing for the genetic mutation identified in the index case (Fig 1A).

## Diagnostic mitochondrial investigations of patient muscle

Postmortem tissue from the patient was made available for diagnostic evaluation of suspected mitochondrial disease. Histopathological assessment, including routine histology and oxidative enzyme histochemistry, showed no obvious morphological abnormalities although the individual COX staining was weak, which was confirmed by a generalised decrease in COX activity following sequential COX-SDH histochemistry (Fig 1B). In agreement with these observations, the assessment of mitochondrial respiratory chain complex activities in skeletal muscle showed decreased activity of complex IV, with activities of complexes I-III within the normal range (Fig 1C).

## Molecular genetic investigations

A prominent complex IV deficiency in patient skeletal muscle demonstrated a mitochondrial aetiology. Assessment of mtDNA in patient samples showed no mutations, and mtDNA copy number was normal. Candidate genes including *SURF1*, *SCO1*, *SCO2*, *COX10*, *COX14*, *COX15*, *COX20*, *COA3*, *COA5* and *APOPT1* were screened but no pathogenic variants were identified. Further analysis using WES revealed biallelic variants in *OXA1L* (NM_005015.3) c.500_507dup, p.(Ser170Glnfs*18) (ClinVar: SCV000803663) and c.620G>T, p.(Cys207Phe) (ClinVar: SCV000803664). The c.500_507dup variant is not listed in the gnomAD database (http://gnomad.broadinstitute. org), whereas the c.620G>T variant is present in one individual of East Asian ethnicity (1 in 246,200, allele frequency 0.00041%). Sanger sequencing confirmed the presence of the variants and that the variants segregated with disease in the family (Fig 1D and E). The c.620G>T *OXA1L* variant is predicted to result in a p.(Cys207Phe) amino acid substitution, but is also predicted to affect splicing due to this variant being at the first nucleotide of the exon. Analysis of cDNA confirmed a splicing defect with evidence of skipping of exon 4 (Fig 1F) demonstrating the c.620G>T variant causes an amino acid substitution and exon skipping (p.[Cys207Phe, Cys207_Glu254del]),

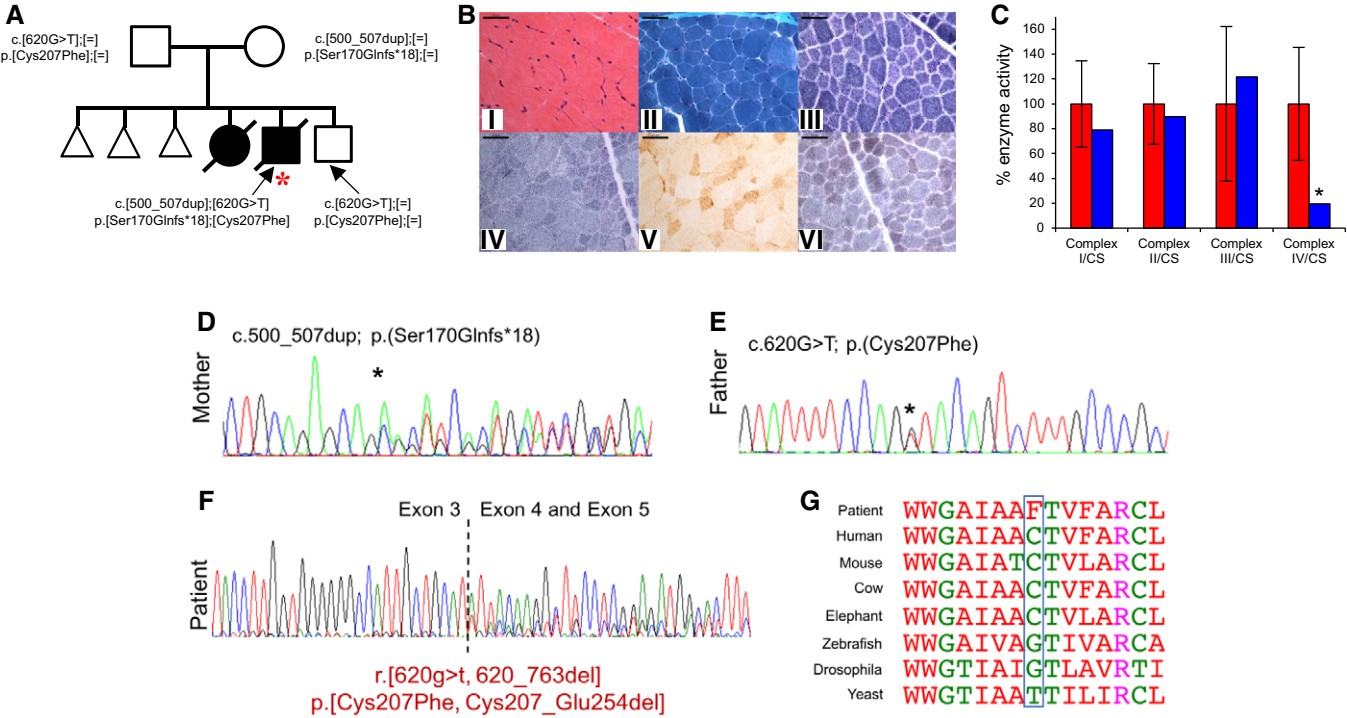

**Figure 1. Molecular genetics and biochemical studies of *OXA1L* variants.**

A   Family pedigree detailing recessive inheritance pattern of *OXA1L* variants, index case denoted with red asterisk.

B   Haematoxylin & Eosin (i) and modified Gomori trichrome (ii) staining demonstrate expected variability in muscle fibre size, with isolated internal nuclei. There is no evidence of regenerative fibres or necrosis, nor subsarcolemmal mitochondrial aggregates typical of "ragged-red" changes. The absence of mitochondrial proliferation is confirmed in both the NADH-tetrazolium reductase (iii) and SDH (iv) reactions. The individual COX reaction (v) reveals a uniform loss of enzyme reactivity across the muscle cryosection, accentuated in the sequential COX-SDH reaction (vi) in which COX-deficient, SDH-positive fibres are prominent uniformly. Scale bar shown is 50 μm.

C   Activity of mitochondrial respiratory complexes in control (red) and patient (blue) skeletal muscle samples. Mean enzyme activities of control muscle (n = 25) are set to 100%, and error bars represent standard deviation. Asterisk represents values outside of the control range.

D, E   Confirmatory Sanger sequencing to show each parent was heterozygous for a single *OXA1L* variant. Asterisk indicates position of the labelled variant.

F   Analysis of mRNA from the patient by RT–PCR showed that the c.620G>T variant affects splicing and can lead to skipping of exon 4 (p.(Cys207_Glu2545del)) or the p.Cys207Phe amino acid substitution.

G   Evolutionary conservation of OXA1L human Cys207 residue (blue box).

with the amino acid substitution affecting the moderately well-conserved Cys207 residue (Fig 1G). Since one variant causes a frameshift and the other affects splicing, American College of Medical Genetics and Genomics (ACMGG) guidelines consider these to be loss-of-function alleles. MutationTaster (www.mutationtaster.org) predicts the c.620G>T variant to be disease-causing, and the variant has a scaled CADD score of 24.9 (http://cadd.gs.washington.edu).

## Neuropathology

The pons exhibited bilateral cavitation with microvascular proliferation, which severely affected the pedunculopontine (PPN) nucleus (Fig EV2Ai, ii and iii) and was accompanied by astrogliosis (Fig EV2Aiv) and microglial activation (Fig EV2Av) with preserved neuronal cell density. PPN neurons and the surrounding neuropil showed profound loss in the levels of complex I subunits NDUFB8 and NDUFS3 expression (Fig EV2Avi), while nuclear-encoded complex II subunit SDHA (Fig EV2Aviii), mitochondrially encoded complex IV subunit COXI (Fig EV2Ax) and complex V subunit ATP5B (Fig EV2Axii) levels were maintained. Substantia nigra neurons

showed a similar pattern of NDUFS3-specific loss (Fig EV2Bi), while SDHA (Fig EV2Bii), COXI (Fig EV2Biii) and ATP5B (Fig EV2Biv) were preserved. The spinal cord demonstrates prominent demyelination of the fasciculus gracile tract (Fig EV2Ci and ii) with relative preservation of myelin in the fasciculus cuneatus (Fig EV2Ciii). The motor neurons were well populated with remaining neurons exhibiting reduced levels of NDUFB8 (Fig EV2Civ) and NDUFS3 (Fig EV2Cv), while SDHA (Fig EV2Cvi), COXI (Fig EV2Cvii) and ATP5B (Fig EV2Cviii) were apparently normal within neurons. Macroscopically, the frontal lobe was atrophic, and the corpus callosum was affected by hypoplasia, particularly in the splenium (Fig EV3A). The basal ganglia were also affected by microcavitation affecting the thalamus (Fig EV3B), and striatum with microvascular proliferation, activated microglia and reactive gliosis (Fig EV3B). The cortex revealed a normal laminar architecture, with discrete cell loss affecting the frontal cortex. Slight downregulation in the levels of NDUFS3 and NDUFB8 relative to SDHA, COXI and ATP5B was observed in neurons within the frontal (Fig EV3C), parietal (Fig EV3D) and occipital cortex (Fig EV3E). The architecture of the cerebellum was preserved with normal Purkinje cell density, minimal

neuronal cell loss from the dentate nucleus and intact expression of NDUFS3, NDUFB8, SDHA, COXI and ATP5B.

## OXPHOS steady-state levels and complex assembly in fibroblasts and skeletal muscle

Western blot analysis of patient fibroblasts and skeletal muscle was carried out to assess the steady-state levels of OXA1L and OXPHOS components. OXA1L protein levels were decreased both in patient fibroblasts (Fig 2A) and in skeletal muscle (Fig 2B) compared to controls demonstrating a functional consequence of the identified *OXA1L* variants. Interestingly, there was no observable difference in the size of the OXA1L protein in the patient sample, suggesting that the protein missing exon 4 is either not produced or rapidly degraded. This would infer that the residual levels of OXA1L in the patient are likely to be the variant carrying

the p.(Cys207Phe) amino acid substitution only. Western blot assessment of subunits of the OXPHOS complexes showed a decrease of the mtDNA-encoded complex IV subunits (COXI and COXII) both in fibroblasts (Fig 2A) and in skeletal muscle (Fig 2B), which is consistent with the deficiency in complex IV activity in skeletal muscle (Fig 1C) and strongly supports a role of OXA1L in the assembly of complex IV. Patient fibroblasts (Fig 2A) and skeletal muscle (Fig 2B) also showed a decrease in the steady-state levels of complex V subunit ATP5B and a slight decrease in the complex I subunit NDUFB8 on Western blots, consistent with previously reported complex V and complex I defects in OXA1L-depleted HEK293 cells (Stiburek *et al*, 2007). Furthermore, complexes I, IV and V assembly defects were observed in patient fibroblasts (Fig 2C) and skeletal muscle (Fig 2D) when assessed by Blue Native PAGE. There appear to be some tissue-specific differences, for example, the levels of fully assembled complex I

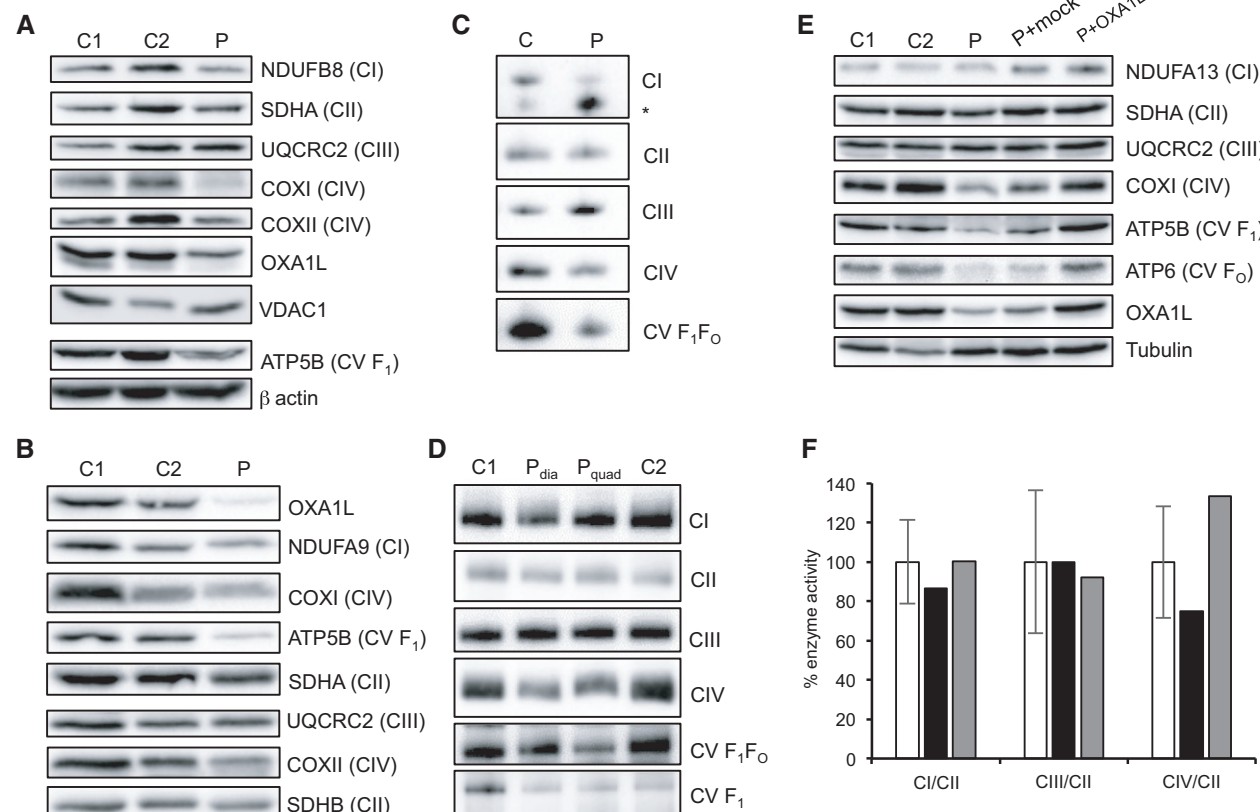

**Figure 2. OXPHOS complex steady-state levels and assembly analysis in patient tissues.**

A Western blot analysis of OXA1L and OXPHOS complex subunits in patient (P) and control (C1, C2) fibroblasts using VDAC1 and β-actin as loading controls.

B Western blot analysis of OXA1L and OXPHOS complex subunits in patient and control skeletal muscle. Complex II subunits SDHA and SDHB were used as loading controls.

C Analysis of respiratory complex assembly by Blue Native PAGE in patient fibroblasts solubilised using DDM as a detergent. Antibodies against NDUFB8 (CI), SDHA (CII), UQCRC2 (CIII), COXI (CIV) and ATP5A (CV) were used. * denotes subassembly complex.

D Analysis of respiratory complex assembly by Blue Native PAGE in patient skeletal muscle from the diaphragm ($P_{dia}$) and quadriceps ($P_{quad}$). Muscle mitochondrial extracts were solubilised using DDM, and antibodies against NDUFB8 (CI), SDHA (CII), UQCRC2 (CIII), COXI (CIV) and ATP5A (CV) were used to detect each complex.

E Western blot analysis of control fibroblasts (C1, C2), patient fibroblasts (P), patient fibroblasts mock transduced with retroviral vector (P+Mock) and patient fibroblasts transduced with retroviral vector containing wild-type human *OXA1L* (P+OXA1L). Alpha tubulin was used as a loading control.

F Activities of mitochondrial respiratory complexes I (CI), III (CIII) and IV (CIV), normalised to complex II (CII), in control fibroblasts (white), patient fibroblasts (black) and patient fibroblasts transduced with wild-type human *OXA1L* (grey). Mean enzyme activities of control fibroblasts (n = 11) are set to 100%, and error bars represent standard deviation.

Source data are available online for this figure.

(980 kDa) was markedly lower in patient fibroblasts compared to controls, but there is an increase in a subassembly complex (Fig 2C). This subassembly intermediate is not present in skeletal muscle, and the decrease in fully assembled complex I in patient skeletal muscle is much less pronounced than in fibroblasts (Fig 2D). In contrast, the assembly of complex IV is markedly decreased both in fibroblasts (Fig 2C) and in skeletal muscle (Fig 2D) and is consistent with the seemingly isolated complex IV deficiency initially seen in skeletal muscle (Fig 1C).

### Retroviral transfection of patient fibroblasts with wild-type *OXA1L* rescues the OXPHOS defect

In order to confirm that the identified variants in *OXA1L* were pathogenic in the patient, we cloned the full-length human *OXA1L* cDNA into a retroviral vector and transduced patient fibroblasts. After viral transduction, OXA1L protein levels in patient fibroblasts increased to levels similar to endogenous OXA1L in control fibroblasts (Fig 2E). Steady-state levels of complex IV and complex V subunits were also increased to control levels by the introduction of wild-type *OXA1L* into patient fibroblasts (Fig 2E). Activities of respiratory complexes were also assessed in patient fibroblasts, showing an increase in complex IV activity in patient fibroblasts transduced with wild-type OXA1L (Fig 2F). These data demonstrate that the OXPHOS defects seen in patient fibroblasts were caused by the *OXA1L* variants, confirming pathogenicity.

### OXA1L depletion leads to a combined OXPHOS defect, including loss of complex IV

Since we observed tissue-specific OXPHOS defects—i.e. a more prominent complex IV defect in skeletal muscle, a seemingly isolated CI deficiency in brain tissue and a combined respiratory deficiency in fibroblasts—we further investigated the importance of OXA1L by a variety of independent means.

Firstly, OXA1L was depleted in *Drosophila* to assess potential effects on OXPHOS complexes using an RNAi construct against the predicted OXA1L *Drosophila* orthologue gene: *CG6404*. Depletion of *CG6404* was confirmed at the mRNA level by qPCR (Fig 3A) as the human OXA1L antibody did not recognise the fly orthologue, and there is no antibody against the fly protein. Steady-state levels of representative subunits from all OXPHOS complexes were decreased in OXA1L-depleted flies (whole fly lysates) compared to controls, with complex I (NDUFS3) being the most affected (Fig 3B). However, porin levels were not different between mutant and control flies indicating that *CG6404*-depleted flies do not have less mitochondrial mass (Fig 3B). OXPHOS deficiency was confirmed by high-resolution respirometry that, as expected, showed a marked reduction of oxygen consumption in OXA1L-IR flies (Fig EV4).

Secondly, we investigated the effects of artificially reducing OXA1L in human cell lines (U2OS) using siRNA. A combined Smart-Pool of four different *OXA1L* siRNAs led to efficient depletion of OXA1L and markedly decreased the steady-state levels of complex IV subunit COXII, demonstrating a clear complex IV defect in U2OS cells depleted of OXA1L (Fig 3C). Furthermore, the levels of several mitoribosomal proteins (MRPs) were also decreased in the OXA1L-depleted cells (Fig 3C). The same observations were made when using two of the OXA1L siRNAs in isolation, with levels of COXII,

MRPL45 and MRPS26 decreasing in line with levels of OXA1L protein (Fig 3D).

In a final approach, we sought to knockout the expression of *OXA1L* in HEK293T cells using CRISPR/Cas9. Following a number of attempts using different guide RNAs (gRNAs), we were unable to identify clones with a complete disruption in all *OXA1L* alleles. This is consistent with OXA1L being found to be an essential fitness gene in a genome-wide CRISPR screen (Hart *et al*, 2015). We therefore undertook an approach previously used to deplete an essential gene (Stroud *et al*, 2016). We generated expression constructs harbouring silent base-pair changes in the *OXA1L* cDNA allowing the amino acids sequence to remain unchanged but that would block both binding of the gRNAs and cutting by our CRISPR/Cas9 constructs. Two variants were made for separate gRNA targets. Both variants were cloned into a lentiviral expression vector with a C-terminal FLAG tag downstream of a doxycycline-inducible promoter, before stable transduction into HEK293T cells. These cells were then subjected to targeting of the *OXA1L* genomic alleles using the appropriate CRISPR/Cas9 construct in the presence of doxycycline. Single cells were sorted and analysed for loss of endogenous OXA1L by immunoblotting. Two cell lines, from different gRNA targeting approaches (termed OXA1L-1 and OXA1L-2), were selected and subsequently characterised. Cells incubated with increasing levels of doxycycline were assessed for OXA1L-FLAG expression by Western blot analysis (Fig 3E). Doxycycline was subsequently used at concentrations of 10 or 20 ng/ml for OXA1L-1 and OXA1L-2, respectively. Cells cultured in the absence of doxycycline succumbed after 7 days, suggesting expression of OXA1L is essential for cell viability (data not shown). Next, cells were incubated in the absence or presence of doxycycline for 5 days before mitochondria were isolated and analysed by BN-PAGE for changes in the levels of OXPHOS complexes harbouring mtDNA-encoded subunits (Fig 3F). As can be seen, depletion of OXA1L led to clear and substantial loss in OXPHOS complexes I, III, IV and V. In addition, loss of OXA1L also led to decreased levels of mitoribosomal subunits (Fig 3G), consistent with OXA1L depletions using siRNA in U2OS cells (Fig 3C and D).

### OXA1L-FLAG immunoprecipitation enriches mtDNA-encoded proteins and assembly factors of complexes I and IV

Next, we asked what proteins associate in complexes with OXA1L. Mitochondria from OXA1L-1 cells cultured in the presence of doxycycline were solubilised in digitonin before affinity enrichment of OXA1L-FLAG using anti-FLAG beads. Bound material was eluted with FLAG peptide and subjected to quantitative mass spectrometry and compared against control HEK293T cells subjected to the same treatment. As seen in Fig 4A, nine of the 13 mtDNA-encoded subunits from complexes I, IV and V were enriched with OXA1L-FLAG. Many nuclear-encoded subunits together with known assembly factors were also detected (Dataset EV1), indicating that assembly intermediates may be assembled during the insertion of core subunits into the inner membrane.

### Assessment of mitochondrial translation and mitoribosomal proteins in patient fibroblasts

As several MRPs were decreased in our siRNA and CRISPR OXA1L knockdowns, we next assessed mitochondrial protein

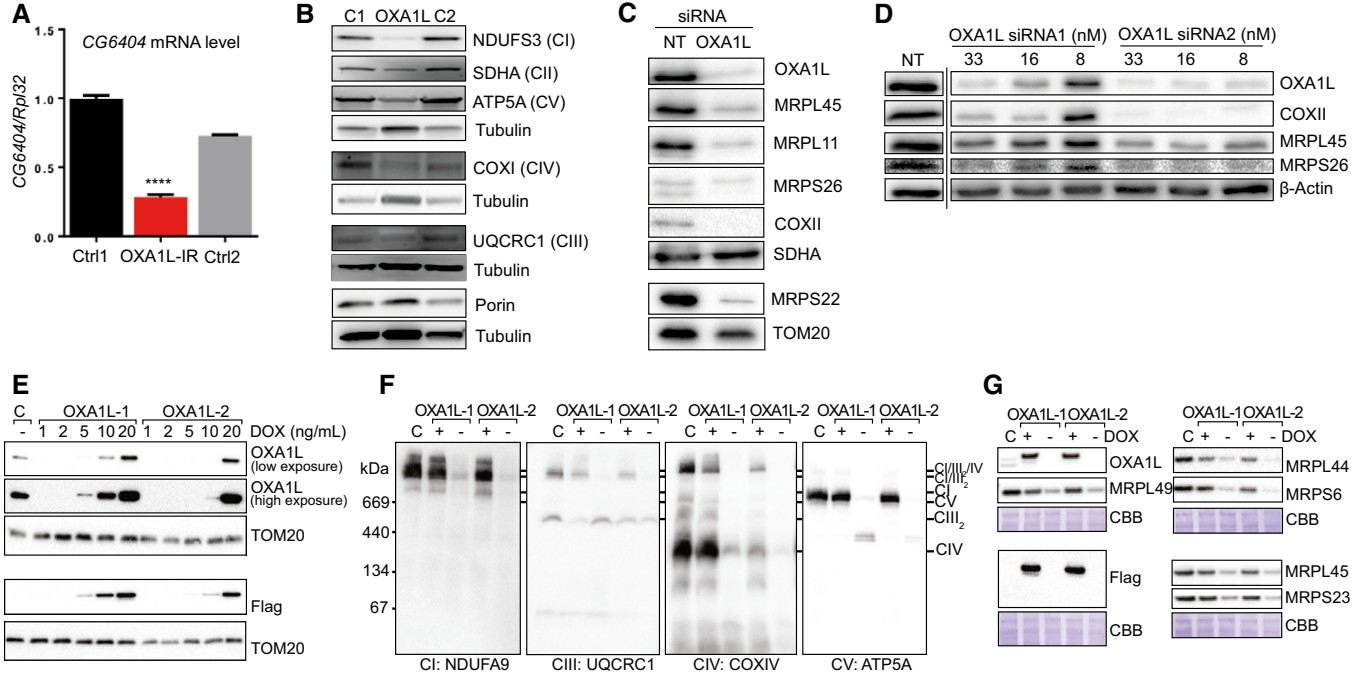

**Figure 3.  Analysis of OXA1L depletion in *D. Melanogaster* and human cells with siRNA and CRISPR/Cas9.**

A   *CG6404* mRNA level in control (black and grey) and OXA1L siRNA-treated flies (red) normalised to RpL32 mRNA with control 1 (black) set to 1. Four biological replicates per group. One-way ANOVA with Tukey's multiple comparison test. Error bars represent standard error of the mean (SEM). **** denotes *P* < 0.0001.

B   Western blot analysis of OXPHOS complex subunits in fly homogenates from control (Ctrl 1, Ctrl 2) and *CG6404* siRNA-depleted (OXA1L-IR) flies. Three biological replicates per group. One-way ANOVA with Tukey's multiple comparison test.

C   Representative figure of Western blot analysis performed on cell lysate (≈30 μg) after 6 days of incubation with either NT-siRNA or OXA1L SmartPool siRNA. The steady-state level of OXA1L and of components of the mt-LSU (MRPL45, MRPL11) and mt-SSU (MRPS26, MRPS22) were evaluated, as well as the steady-state level of the mtDNA-encoded protein COXII. SDHA and TOM20 antibodies were used as loading controls.

D   Western blot analysis was performed on cell lysate (≈30 μg). The efficiency of the depletion was assessed with antibodies targeting OXA1L. Steady-state levels of the mt-encoded protein COXII as well as mitoribosomal proteins MRPL45 and MRPS26 were also visualised. The equality of the loading was controlled using antibodies targeting β-actin. The figures are representative of two biological repeats. The dashed line indicates that some lanes were omitted from the figure.

E   Two independent cell lines (OXA1L-1 and OXA1L-2) were grown in different concentrations of doxycycline (DOX) to modulate OXA1L^FLAG expression. Mitochondria were isolated and compared with control mitochondrial from wild-type HEK293T cells (labelled C) by SDS–PAGE and immunoblotting with antibodies against OXA1L, FLAG epitope and TOM20 (as control).

F   Mitochondria were isolated from cells treated with or without 10 ng/ml DOX for 5 days. Samples were solubilised in digitonin before BN-PAGE analysis of supercomplex forms of respiratory complex I (NDUFA9), complex III (UQCRC1), complex IV (COXIV) and complex V (ATP5A) by immunoblotting.

G   Ribosomal subunits were analysed by SDS–PAGE and immunoblotting. Blots stained with Coomassie brilliant blue (CBB) serve a loading controls.

Source data are available online for this figure.

synthesis in patient fibroblasts. Western blotting analysis revealed no change in the steady-state protein levels of MRPL3, MRPL11, MRPL45 or MRPS26 in patient fibroblasts compared to control (Fig 4B). The stability of the mitoribosome in patient fibroblasts may be explained by the higher residual levels of OXA1L (approximately 30% relative to control fibroblasts) than in either of the knockdown experiments, where residual OXA1L levels are much lower. To assess *de novo* mitochondrial protein synthesis in human fibroblasts, cells were incubated with media lacking methionine and cysteine while concomitantly inhibiting cytosolic translation with emetine dihydrochloride. Cells were then incubated with [$^{35}$S] labelled methionine and cysteine for 1 h (pulse) followed by either a 4-h or 8-h period with normal growth media containing methionine (chase). There was no clear difference in *de novo* mitochondrial protein synthesis between control and patient fibroblasts in the 1 h pulse, but after 8 h chase, there was a marked decrease in signal from the newly synthesised

mitochondrial proteins in the patient fibroblasts (Fig 4C). These data suggest that the mitoribosomes are translating efficiently in patient fibroblasts, but that the nascent polypeptides are less stable and degrade more rapidly than in control cells. These data are consistent with the role of OXA1L in the co-translational insertion of mtDNA-encoded proteins into the IMM (Szyrach *et al*, 2003; Haque *et al*, 2010).

# Discussion

Here, we describe a case of mitochondrial disease caused by mutations in *OXA1L*, which leads to a severe encephalopathy, hypotonia and developmental delay. We have established the pathogenicity of these variants by showing rescue of the CI, CIV and CV defects in patient fibroblasts after introducing wild-type *OXA1L* via a retroviral vector (Fig 2E).

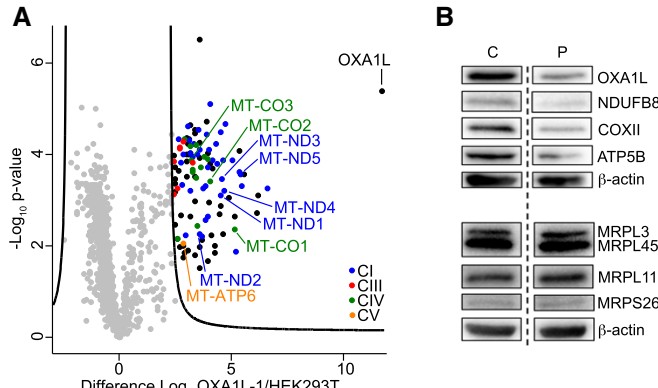

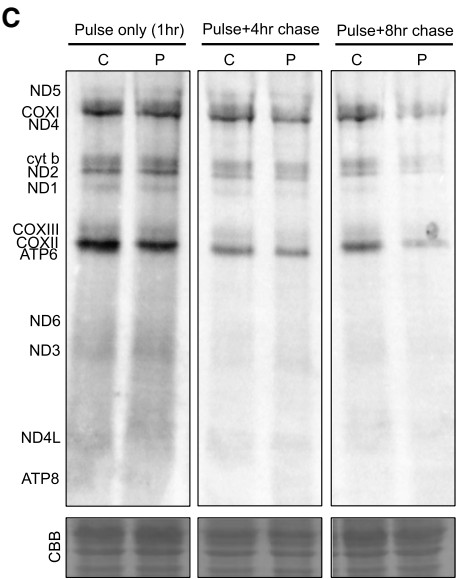

**Figure 4.   Identifying interacting partners of OXA1L and assessing mitochondrial protein synthesis in patient fibroblasts.**

A   Affinity purification mass spectrometry analysis of proteins interacting with OXA1L-FLAG. Control (HEK293T) or OXA1L-1 cells expressing OXA1L-FLAG were solubilised in 1% (w/v) digitonin and incubated with anti-FLAG affinity gel ($n = 3$). Eluates were analysed by label-free quantitative mass spectrometry (LFQ). Log$_2$ LFQ intensities were submitted to a modified two-sided two-sample $t$-test with significance determined through permutation-based false discovery rate (FDR) statistics as described in the Materials and Methods. Enriched respiratory complex subunits are colour-coded, and mtDNA-encoded subunits are labelled. Full data are provided in Dataset EV1.

B   Western blot analysis was performed from control (C) and OXA1L patient immortalised fibroblasts (P). The steady-state level of OXA1L was assessed as were the steady-state levels of components of the OXPHOS system (NDUFB8, COXII and ATP5β). On a separate Western blot, the levels of the mitoribosomal proteins MRPL3, MRPL45, MRPL11 and MRPS26 were detected. For both the membranes, β-actin was used as loading control. The dashed line indicates that some lanes were omitted from the figure.

C   Control (C1) and patient (P) fibroblasts were treated with emetine dihydrochloride to inhibit cytosolic translation and mitochondrial protein synthesis analysed by [$^{35}$S] Met/Cys incorporation with 1 h pulse and either 4 or 8 h chase. Cell lysate (50 μg) was separated through a 15% polyacrylamide gel. The gel was stained with Coomassie blue (CBB) to confirm equal loading. After fixation and drying the signal was visualised using a Typhoon FLA9500 PhosphorImager. Signals were ascribed following established migration patterns (Chomyn, 1996).

Source data are available online for this figure.

Interestingly, initial diagnostic tests showed a seemingly isolated complex IV deficiency in patient skeletal muscle, prompting the sequencing of several COX assembly genes. However, subsequent neuropathology experiments indicated an isolated complex I deficiency in the central nervous system. It is also worth noting that while patient skeletal muscle initially appeared to show an isolated complex IV deficiency in terms of activity (Fig 1C), Western blotting and BN-PAGE showed a clear complex V defect (which was not assessed by enzymatic assay) and milder defects in complex I (Fig 2C and D) in addition to the complex IV defect. The tissue-specific effects of the OXA1L variants remain unexplained, but perhaps suggest that the mitochondrial insertase machinery can vary between tissues, possibly due to differential expression of OXA1L isoforms or that other, as yet unidentified, insertases present in human mitochondria can substitute for OXA1L in different tissues. Interestingly, available mRNA expression data for OXA1L suggest that brain tissue has the lowest relative expression of OXA1L and that the expression pattern is fairly similar between the various isoforms (Fig EV5). Recently, more distant homologues of Oxa1 have been identified in the eukaryotic endoplasmic reticulum, namely the DUF106-related proteins WRB/Get1, EMC3 and TMCO1 (Anghel et al, 2017). Similar phylogenetic analysis yielded the same list of homologues with no additional insertase candidates (Appendix Fig S1 and Appendix Table S2).

To our knowledge, the only previous report of OXA1L function in humans was conducted with shRNA depletions in HEK293 cells (Stiburek et al, 2007) and demonstrated clear defects in complexes I and V, but complex IV remained unchanged. This was surprising as OXA1L was initially identified as a human gene involved in complex IV assembly through its functional complementation of a complex IV defect in Oxa1p knockout yeast strains (Bonnefoy et al, 1994b). We saw a similar sparing of complex IV in the brain tissue of the patient, with only an isolated complex I defect detected through neuropathology studies (Figs EV2 and EV3), but patient fibroblasts and skeletal muscle showed clear complex IV defects. We sought to corroborate our findings in patient tissues and to demonstrate the effects of OXA1L depletion on complex IV assembly by other means. We saw clear defects in complex IV, as well as complexes I and V when OXA1L was depleted using: (i) siRNA in D. melanogaster (Fig 3B), (ii) siRNA in human U2OS lines (Fig 3C), and (iii) CRISPR/Cas9 in HEK293T cells (Fig 3F). Taken together, these data demonstrate that OXA1L is indeed important for complex IV assembly in many human cell types and can act as a general insertase for the majority of mtDNA-encoded subunits into the inner mitochondrial membrane. Similarly, we observed that all respiratory complexes including complex IV were affected by depletion of CG6404 (the fly orthologue of OXA1L) in Drosophila. Mitochondrial respiration was strongly affected (Fig EV4) and flies died shortly (1 or 2 days) after eclosion. These data support an essential role for OXA1L in the assembly of respiratory complexes that have been conserved during evolution.

Immunoprecipitation experiments showed that OXA1L-FLAG interacts with at least nine of 13 mtDNA-encoded proteins along with many nuclear-encoded subunits and assembly factors of complexes I, III, IV and V (Fig 4A, Dataset EV1), suggesting OXA1L may have a role in assembly of these complexes around the co-translational insertion of the core subunits. We have also shown that mitochondrial protein synthesis is not affected in patient

fibroblasts, but that the stability of nascent polypeptides is decreased, which is also consistent with a role for OXA1L in co-translational insertion and respiratory complex assembly.

In both siRNA and CRISPR/Cas9 OXA1L knockdown experiments, there was a loss of mitoribosomal proteins (Fig 3C, D and G), which may be a secondary effect of severe OXA1L depletion since the patient cells with a milder OXA1L depletion do not show loss of MRPs or impaired mitochondrial translation (Fig 4B and C). It is likely that the extreme knockdown of OXA1L is causing loss of MRPs as a downstream secondary effect, possibly due to problems with the mitochondrial membrane potential and/or protein import. This is consistent with the fact that several members of the protein import machinery and certain mitochondrial carriers were also shown to interact with OXA1L in the OXA1L-FLAG immunoprecipitation experiments (Dataset EV1). Problems with protein import would also account for the decreased SDHA seen in OXA1L-depleted flies (Fig 3B). In yeast, Oxa1p has been shown to be required for the biogenesis of the Tim18-Sdh3 module of the TIM22 complex (Stiller et al, 2016), which is responsible for the import of many inner membrane proteins. This demonstrates that, in yeast, Oxa1p plays a major role in the biogenesis of other inner membrane proteins as a secondary consequence of impaired protein import in Oxa1-deficient mitochondria. The fact that MRP levels are not affected in the patient suggests that the primary effects of OXA1L are in the co-translational insertion of mtDNA-encoded proteins and OXPHOS complex assembly. It is likely that complete loss-of-function mutations are not seen in patients due to more severe mutations not being compatible with life, which is supported by the CRISPR/Cas9 OXA1L knockouts only surviving for 7 days without doxycycline treatment and the death of OXA1L-depleted flies shortly after eclosion.

In yeast, Oxa1p has long been known to be required for the correct insertion of several mtDNA-encoded subunits (Hell et al, 1997, 2001; Jia et al, 2007). Furthermore, Oxa1p plays a crucial role in N-tail protein export in mitochondria, where the N-terminal tail of inner membrane proteins is translocated from the mitochondrial matrix to the intermembrane space (Hell et al, 1998). This process applies to nuclear-encoded mitochondrial inner membrane proteins that are imported to the matrix and also to some of the mtDNA-encoded proteins that require the N-terminal tail to reside on the intermembrane space side of the membrane, such as Cox2p. The C-terminal tail of Cox2p is also translocated across the membrane to the intermembrane space, and this process is also impaired in yeast strains lacking Oxa1p (He & Fox, 1997). Another member of the Oxa1 insertase family, Cox18p, is also essential for the translocation of the C-terminal tail of Cox2p in yeast (Saracco & Fox, 2002). Recently, it has been shown that knockout of COX18 (the human homologue of Cox18p) using TALENs impairs the translocation of the C-terminal tail of COXII in human cell lines (Bourens & Barrientos, 2017). This demonstrates a strong functional conservation in the insertion of COXII between yeast and humans. Knockout of COX18 had no effect on the N-terminal insertion of COXII, suggesting other insertases are involved in this process. Since Oxa1p is essential for N-terminal insertion of Cox2p in yeast (Hell et al, 1998) and the homologues Cox18p and COX18 are functional orthologs, it is likely that OXA1L is at least partially responsible for N-terminal insertion of COXII in humans. Our data are consistent with this notion as COXII is decreased in patient fibroblasts (Fig 2A), skeletal muscle (Fig 2B) and OXA1L

siRNA-depleted U2OS cells (Fig 3D), and COXII was significantly enriched in the OXA1L-FLAG immunoprecipitation experiments (Fig 4A). OXA1L is not only important for COXII insertion, but is more likely to be a more generalised insertase similarly to Oxa1p in yeast (Hell et al, 2001).

In summary, our data confirm the pathogenicity of the c.500_507dup, p.(Ser170Glnfs*18) and c.620G>T, p.(Cys207Phe) OXA1L variants and demonstrate a role for OXA1L in the co-translational insertion/assembly of OXPHOS complexes I, IV and V.

# Materials and Methods

## Neuropathology

Postmortem CNS tissues were taken from a 5-year-old male and were fixed in formalin for 21 days and paraffin-embedded before undergoing routine neuropathological examination. In order to document the severity of the pathological changes in patient tissues, CNS tissues from two non-neurologically affected controls were obtained from the Edinburgh Brain and Tissue Bank. Histology was performed as previously described (Lax et al, 2012).

## Whole exome sequencing

Whole exome sequencing was undertaken using an Illumina paired-end pre-capture library and hybridised to the HGSC CORE exome capture design (52 Mb, NimbleGen) and subsequently sequenced on a HiSeq 2500 to an average 100× average coverage. Bioinformatic analyses were conducted as previously described (Stiles et al, 2015). Single nucleotide variants (SNVs) and small insertions and deletions (InDels) were scored by GATK (DePristo et al, 2011). Quality control filtering of variants was based on coverage, strand bias, mapping quality and base quality. Custom Perl scripts were used to annotate variants as previously described (Bonnen et al, 2013; Stiles et al, 2015). Algorithms used for prediction of potential functional consequences of variants included CADD (Kircher et al, 2014), SIFT (Ng & Henikoff, 2002) and PolyPhen2 (Adzhubei et al, 2013), Genomic Evolutionary Rate Profiling (GERP; Davydov et al, 2010), and PhyloP (Cooper et al, 2005).

## Generation of the OXA1L depletion line

HEK293T cells were cultured in DMEM supplemented with 10% FBS, penicillin/streptomycin and 50 μg/ml uridine (Sigma). Generation of OXA1L-inducible knockout cell lines was performed by disrupting the genomic OXA1L copies and inserting OXA1L into cells under a doxycycline-inducible promoter as previously described for NDUFAB1 (Stroud et al, 2016). Briefly, two different doxycycline-inducible OXA1L constructs containing silent mutations to either destroy the PAM site or block gRNA recognition along with a C-terminal FLAG epitope (referred to as OXA1L-FLAG) were cloned into the pLVX-TetOne-Puro plasmid (Clontech) and transduced into clonal HEK293T cells using lentivirus particles. Oxa1L-1 CRISPR/Cas9 targeted exon 2 using the guide sequence 5′-GCGAGGGCCCTGCGACGCTG-3′ and the cDNA stably transduced harboured the silent mutation c.60 C > G to mutate the PAM site.

Oxa1L-2 targeted exon 3 using the guide sequence 5′-GATCTGGGCCTACCTTGGTG-3′ and the cDNA stably transduced harboured the silent mutations (5′-GACCTCGGGTTGCCATGGTG-3′) to mutate the gRNA-binding site (as the PAM site could not be mutated in this case). Cells were grown in the presence of 2 μg/ml puromycin for 72 h to select transduced cells, and expression of OXA1L-FLAG was confirmed after expression with 1 μg/ml doxycycline (DOX; Clontech) for a further 72 h, followed by SDS–PAGE and immunoblotting with OXA1L (Proteintech) or FLAG (Sigma-Aldrich) antibodies. Once OXA1L-FLAG expression was confirmed, cells were grown in the presence of 50 ng/ml DOX and then transfected with pSpCas9(BB)-2A-GFP (Addgene 48138, a gift from F. Zhang; Ran *et al*, 2013) encoding the appropriate gRNA sequence. Cells were single-cell FACS-sorted, and clones were selected based on loss of endogenous OXA1L. Cells were maintained in the presence of 50 ng/ml DOX with media being changed every 48 h. Loss of endogenous OXA1L was confirmed by performing Western blotting of cell extracts and immunodecoration with OXA1L antibodies and sequencing of alleles for indels. Doxycycline concentrations were titrated to match expression levels with endogenous OXA1L in control cells.

## Analysis of human OXA1L depletion

For OXA1L depletion experiments, cells were cultured in DMEM containing 10% dialysed FBS (Sigma) and grown for 5 days in the presence or absence of 10 ng/ml DOX. Mitochondria were isolated and proteins and their complexes analysed by BN-PAGE as previously described (Formosa *et al*, 2015) and NuPAGE SDS–PAGE (Thermo Fisher).

## Affinity purification mass spectrometry (AP-MS)

Affinity purification mass spectrometry was performed using whole cell lysates. For affinity purification, control HEK293T and OXA1L-FLAG cells (3 mg total protein) were resuspended in triplicate in 500 μl AP buffer (20 mM Bis-Tris pH 7, 50 mM NaCl, 10% (v/v) glycerol) containing 1% digitonin and incubated for 30 min on ice. Insoluble material was removed by centrifugation at 21,000 *g*, and the supernatant was incubated with 30 μl anti-FLAG M2 affinity gel (Sigma) for 1 h with rotation at 4°C. Beads were washed 10 times with AP buffer containing 0.1% (w/v) digitonin and finally eluted with 200 μl of AP buffer containing 150 μg/ml FLAG peptide (Sigma).

For mass spectrometry, proteins were acetone precipitated, reduced, alkylated, trypsin digested and desalted as previously described (Stroud *et al*, 2016). Peptides were reconstituted in 0.1% (v/v) trifluoroacetic acid (TFA) and 2% (v/v) acetonitrile (ACN) and analysed by online nano-HPLC/electrospray ionisation-MS/MS on a Q Exactive Plus connected to an Ultimate 3000 HPLC (Thermo Fisher Scientific). Peptides were loaded onto a trap column (Acclaim C18 PepMap nano-Trap × 2 cm, 100 μm I.D, 5 μm particle size and 300 Å pore size; Thermo Fisher Scientific) at 15 μl/min for 3 min before switching the pre-column in line with the analytical column (Acclaim RSLC C18 PepMap Acclaim RSLC nanocolumn 75 μm × 50 cm, PepMap100 C18, 3 μm particle size 100 Å pore size; Thermo Fisher Scientific). The separation of peptides was performed at 250 nl/min using a linear ACN gradient

of buffer A [0.1% (v/v) formic acid, 2% (v/v) ACN] and buffer B [0.1% (v/v) formic acid, 80% (v/v) ACN], starting at 12.5% buffer B to 38.5% followed by ramp to 99% over 30 min. Data were collected in positive mode using data-dependent acquisition using m/z 375–1,800 as MS scan range, HCD for MS/MS of the 12 most intense ions with $z \geq 2$. Other instrument parameters were as follows: MS1 scan at 70,000 resolution (at 200 m/z), MS maximum injection time 50 ms, AGC target 3E6, normalised collision energy was at 27% energy, isolation window of 1.8 Da, MS/MS resolution 17,500, MS/MS AGC target of 1E5, MS/MS maximum injection time 100 ms, minimum intensity was set at 1E3 and dynamic exclusion was set to 20 s. Raw files were analysed using the MaxQuant platform (Tyanova *et al*, 2016a) version 1.6.0.1 searching against the UniProt human database containing reviewed, canonical and isoform variants in FASTA format (June 2016) and a database containing common contaminants. Default search parameters for a label-free (LFQ) experiment were used with modifications. Briefly, multiplicity was set to 1 (unlabelled), and "LFQ", "Re-quantify" and "Match between runs" were enabled with default settings. Unique and razor peptides were used for quantification, using a LFQ minimum ratio count of 1. Using the Perseus platform (Tyanova *et al*, 2016b) version 1.5.5.3, proteins identified from < 2 MS/MS were excluded, along with those lacking LFQ intensity values for all three OXA1LFLAG experiments. Missing values for control (HEK293T) experiments were imputed to values based on the distribution of LFQ intensity values in each column, with downshift of 2.2 and width of 0.3. These values were empirically determined to place imputed values on the tail end of intensity distribution. A modified two-sided *t*-test based on permutation-based FDR statistics (Tyanova *et al*, 2016b) was performed between experimental groups, and the negative logarithmic *P*-values were plotted against the differences between the $\text{Log}_2$ means for the two groups. A significance threshold (FDR < 0.01, s0 = 8) was determined that excludes all values on the left (control) side of the test. Raw data, including *q*-values used for indicating significance, can be found in Dataset EV1.

## Fly husbandry

*Drosophila melanogaster* flies were maintained at 25°C for all experiments. Flies were cultured on standard media (1% agar, 1.5% sucrose, 3% glucose, 3.5% dried yeast, 1.5% maize, 1% wheat, 1% soya, 3% treacle, 0.5% propionic acid, 0.1% Nipagin) with a controlled 12 h:12 h light:dark cycle. A line carrying a UAS-RNAi construct against *CG6404* was obtained from the Vienna *Drosophila* Resource Center (*ID:* 108091). The Tubulin-GeneSwitch (tubGS) driver was a generous gift from the laboratory of Dr Scott Pletcher, and virgin females carrying the tubGS driver (backcrossed into our Dahomey control background for six generations) were crossed with (i) males carrying the RNAi construct against *CG6404* or (ii) males from the VDRC control stock $w^{1118}$ to control for the effects of the driver (this group is named Ctrl1 in figures). Additionally, Dahomey virgin females were crossed with males carrying the UAS-RNAi construct against *CG6404* (this group is named Ctrl2 in figures) to control for positional effects of the RNAi construct insertion. Males were collected following eclosion and transferred to new food for one day before being used for experiments.

## Western blot analysis—*Drosophila*

Sample preparation (20 flies were used per sample, three biological replicates per group) and Western blotting were performed as described in Scialò *et al* (2016). The primary antibodies employed together with the appropriate secondary antibodies were as follows: anti-NDUFS3 (Abcam; ab14711) 1:1,000 diluted in 5% milk in PBS-Tween 1x, anti-SDHA (Abcam; ab209986) 1:500 diluted in 5% BSA in PBS-Tween 1x; anti-ATP5α (Abcam; ab14748) 1:100,000 diluted in 5% milk in PBS-Tween 1x, anti-UQCRC1 (Abcam; ab110252) 1:500 diluted in 5% BSA in PBS-Tween 1x, anti-MTCO1 (Abcam; ab14705) 1:500 diluted in 5% BSA in PBS-Tween 1x, Porin (Abcam; ab14734) 1:1,000 diluted in 5% milk in PBS-Tween 1x, anti-beta-tubulin [EPR16774] (Abcam; ab179513) 1:1,000 diluted in 5% milk in PBS-Tween 1x, HRP Horse Anti-Mouse IgG Antibody (Peroxidase; Vector laboratories, California; PI-2000) 1:5,000 diluted in 5% milk, HRP Goat Anti-Rabbit IgG Antibody (Peroxidase; Vector laboratories, California; PI-1000) 1:5,000 diluted in 5% milk. Image quantification: ImageJ was used to quantify the protein band intensities. For each image, the bands were quantified and normalised by the corresponding tubulin intensity used as a loading control.

## High-resolution respirometry—*Drosophila*

$O_2$ consumption was measured in homogenates from whole male flies as described previously (Scialò *et al*, 2016). Briefly, 20 flies were homogenised in assay buffer (120 mM KCl, 5 mM $KH_2PO_4$, 3 mM Hepes, 1 mM EGTA, 1 mM $MgCl_2$, 0.2%, pH 7.2) and incubated in the same buffer containing 0.2% bovine serum albumin at 25°C. Respiration was measured by using an Oroboros O2K (Oroboros Instruments Corp) using specific substrates and inhibitors for complex I (pyruvate/proline and rotenone), complex III (glycerol-3-phosphate, antimycin A) and complex IV (ascobarte/TMPD (N,N,N′,N′-tetramethyl-p-phenylenediamine), cyanide). Values were normalised by citrate synthase activity measured as described previously (Magwere *et al*, 2006). ANOVA test was followed by Tukey's multiple comparisons test. Normality of samples was tested using Shapiro–Wilk test, and Bartlett's test was used to assess homoscedasticity.

## RNA extraction, cDNA synthesis and qPCR—*Drosophila*

RNA extraction was performed as described previously (Scialò *et al*, 2016). Twenty flies were used per sample, four biological replicates per group. cDNA synthesis: cDNA synthesis was carried out using the High-Capacity cDNA Reverse Transcription Kit (Fisher Scientific, Applied Biosystems™; 4368814). qPCR: qPCR was carried out using SYBR green SensiFast Hi-Rox buffer (Bioline: BIO-92020), and the following primers for *CG6404* (forward primer: AGCCCTGCCC ATTGTTATCT, reverse: primer CTTCTTAGGTGGCAGTGCAC) were used. RpL32 was used as internal standard (forward primer: AGGCCCAAGATCGTGAAGAA, reverse primer: TGTGCACCAGG AACTTCTTGAA).

## Retroviral rescue

The full-length human OXA1L cDNA (mammalian genome collection clone BC001669) was amplified by PCR (KAPA HiFi), cloned into pDONR 201 with BR Clonase II (Thermo Fisher) and subsequently cloned into the Gateway converted retroviral vector pBABE-puro with LR Clonase II. Retrovirus was generated by transient transfection of retroviral plasmids into the Phoenix amphotropic packaging line. Transduced fibroblasts were used directly in experiments following selection with puromycin.

## Muscle histology and biochemistry

Informed consent for diagnostic and research-based studies was obtained from the family in accordance with the Declaration of Helsinki protocols and approved by local institutional review boards. Histological and histochemical analyses were performed on 10 µm transversely orientated serial cryosections of a skeletal muscle biopsy sample using standard procedures. The activities of

### The paper explained

**Problem**

Mitochondrial diseases are a group of disorders characterised by mitochondrial dysfunction, predominantly resulting in decreased energy production in the form of ATP made by the oxidative phosphorylation (OXPHOS) system. Mitochondrial diseases are very varied, in terms of both their clinical manifestations and their genetic causes. They can be caused by mutations within the mitochondrial genome itself (mtDNA) or mutations in nuclear genes that encode mitochondrial proteins (≈1,200). More than 300 of these genes have been associated with mitochondrial disease, and this diversity often makes obtaining genetic diagnoses in patients with mitochondrial disease difficult. OXA1L is a mitochondrial protein encoded by the nuclear genome and belongs to the YidC/Alb3/Oxa1 membrane protein insertase family. Oxa1 in yeast is known to be important for the insertion and assembly of subunits of OXHPOS complex IV, but in human cells, OXA1L depletion has been reported to cause defects in complexes I and V only. There have not been any previous reports of *OXA1L* mutations in mitochondrial disease patients.

**Results**

We used whole exome sequencing to identify recessively inherited variants in *OXA1L* (NM_005015.3; c.500_507dup, p.(Ser170Glnfs*18) and c.620G>T, p.(Cys207Phe) in a patient with a mitochondrial disorder, presenting at birth with hypotonia, severe encephalopathy and developmental delay. We showed that OXPHOS complexes I, IV and V were impaired in patient samples with a particularly notable biochemical deficiency of complex IV in skeletal muscle. We also demonstrated a combined OXPHOS defect (including complex IV) with targeted depletion of OXA1L in human cell lines and in a *Drosophila* model. Critically, we confirmed the *OXA1L* variants are causative of the disease in the patient by showing a rescue the OXPHOS defect in patient fibroblasts following the retroviral delivery of a wild-type (normal) copy of *OXA1L*.

**Impact**

Our study documents the first case of mitochondrial disease caused by mutations in OXA1L. Confirmation of these mutations as causative in this case allowed pre-natal testing of a subsequent pregnancy in this family, demonstrating the clinical importance of providing a genetic diagnosis in mitochondrial disease cases. OXA1L may now be screened as a candidate gene in future cases with a similar clinical presentation. Our work also provides clearer insight into the function of OXA1L, given we have shown that OXA1L not only affects complexes I and V, but is also important for the correct assembly and function of complex IV in humans.

individual respiratory chain complexes and citrate synthase, a mitochondrial matrix marker, were determined in muscle homogenates as described previously (Kirby *et al*, 2007).

## Western blotting and Blue Native PAGE from patient tissues

Protein extraction from human skeletal muscle and fibroblast samples for SDS–PAGE and subsequent Western blotting was carried out as described previously (Thompson *et al*, 2016). Blue Native PAGE was conducted on mitochondria isolated from skeletal muscle and fibroblasts samples as described previously (Oláhová *et al*, 2015) using antibodies against COXI (Abcam ab14705), SDHA (Abcam ab14715), Porin/VDAC1 (Abcam ab14734), UQCRC2 (Abcam ab14745), NDUFB8 (Abcam ab110242), ATP5A (Abcam ab14748), ATP5B (Abcam ab14730) and OXA1L (Proteintech 21055-1-AP). All primary antibodies were used at a dilution of 1 in 1,000.

## OXA1L siRNA depletion

siRNA SmartPool (Dharmacon, cat no M-012696-00) and individual siRNA from the SmartPool (Dharmacon, cat no M-012696-03 and M-012696-04) were used for transient depletion of OXA1L in U2OS cells. Cells were incubated for 3 days with either siRNA targeting OXA1L or with non-targeting siRNA, at the final concentration of 33 nM. The siRNA was delivered using Lipofectamine RNAiMAX (Thermo Fisher Scientific) diluted 1:1,000 in Opti-MEM I + GlutaMAX-I (Gibco), as per manufacturer's instructions. After 3 days, adherent cells were re-transfected using the same conditions. At the end of the 6-day depletion, cells were collected to perform Western blot analysis.

## *De novo* mitochondrial protein synthesis

Mitochondrial protein synthesis in cultured cells was performed essentially as described previously (Chomyn, 1996). Fibroblasts were labelled for 1 h in methionine/cysteine-free DMEM (Sigma) with 200 μCi/ml of a [35S]-methionine/cysteine mixture (Perkin Elmer) and 100 μg/ml emetine dihydrochloride (Sigma) followed by either 4 or 8 h chase in standard DMEM with additional 7.5 μg/ml cold methionine. Aliquots (50 μg) of total cell protein were separated by 15% SDS–PAGE. Signals were detected using the Typhoon FLA9500 Phosphorimager and ImageQuant software (GE Healthcare).

## Data availability

The dataset produced in this study is available in the following database:
Clinical data: ClinVar accession numbers: SCV000803663 and SCV000803664 (https://www.ncbi.nlm.nih.gov/clinvar/)

**Expanded View** for this article is available online.

## Acknowledgements

This work is supported by the Wellcome Centre for Mitochondrial Research (203105/Z/16/Z), the Medical Research Council (MRC) Centre for Translational Research in Neuromuscular Disease, Mitochondrial Disease Patient Cohort (UK) (G0800674), the Lily Foundation, The Barbour Foundation, the UK NIHR Biomedical Research Centre for Ageing and Age-related disease award to the Newcastle upon Tyne Foundation Hospitals NHS Trust, the MRC/EPSRC Molecular Pathology Node, the UK NHS Highly Specialised Service for Rare Mitochondrial Disorders of Adults and Children, the National Institute of Neurological Disorders and Stroke of the National Institutes of Health (award number R01NS083726 to PEB), the Biotechnology and Biological Sciences Research Council (BB/M023311/1) and the Australian Mitochondria Disease Foundation (AMDF)). MTR and DAS were funded by the National Health & Medical Research Council (NHMRC) of Australia (Grants APP 1125390, 1107094, 1140906) and a Fellowship (1140851) to DAS. CJ-M is funded by the Instituto de Salud Carlos III and the European Regional Development Fund (FEDER) grant CP09/00011. We thank the "Biobanc de l'Hospital Infantil Sant Joan de Déu per a la Investigació" integrated in the Spanish Biobank Network of ISCIII for patient samples and data procurement.

## Author contributions

Conception and design of the study: KT, MTR, ZMAC-L, RNL and RWT. Acquisition and analysis of data: KT, NM, MO, FS, LEF, DAS, MG, NZL, FMR, CJ, AN, CO, CJ-M, SAH, LH, GKB, PM and PEB. Drafting the manuscript: KT and RWT. Critical revision of the manuscript: NM, MO, FS, LEF, DAS, NZL, CJ, AN, RM, AS, BJB, PEB, MTR, ZMAC-L and RNL. Study supervision: AS, BJB, MTR, ZMAC-L, RNL and RWT.

## Conflict of interest

The authors declare that they have no conflict of interest.

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
