## [Review Process File · EMBO Molecular Medicine]

OXA1L Mutations Cause Mitochondrial encephalopathy and a combined OXIDATIVE PHOSPHORYLATION defect

Kyle Thompson, Nicole Mai, Monika Oláhová, Filippo Scialó, Luke E. Formosa, David A. Stroud, Madeleine Garrett, Nichola Z. Lax, Fiona M. Robertson, Cristina Jou, Andres Nascimento, Carlos Ortez, Cecilia Jimenez-Mallebrera, Steven A. Hardy, Langping He, Garry K. Brown, Paula Marttinen, Robert McFarland, Alberto Sanz, Brendan J. Battersby, Penelope E. Bonnen, Michael T. Ryan, Zofia M.A. Chrzanowska-Lightowlers, Robert N. Lightowlers and Robert W. Taylor.

Review timeline:

Submission date:	1 st March 2018
Editorial Decision:	22 nd March 2018
Revision received:	22 nd June 2018
Editorial Decision:	11 th July 2018
Revision received:	28 th July 2018
Accepted:	14 th August 2018

Editor: Celine Carret

Transaction Report:

1st Editorial Decision

22nd March 2018

Thank you for the submission of your manuscript to EMBO Molecular Medicine. We have now heard back from the three referees whom we asked to evaluate your manuscript.

You will see from the comments of the referees pasted below that they do appreciate the description of a novel disease and the case report, found it important and deserving publication. While referees 3 is supportive, referees 1 and 2 are a little more critical and would like to see more molecular and mechanistic aspects to the paper, improve the quality of the blots and present the clinical data in a manner more accessible to all readers.

We would therefore welcome the submission of a revised version within three months for further consideration and would like to encourage you to address all the criticisms raised as suggested to improve conclusiveness and clarity.

REFeree REPORTS.

Referee #1 (Comments on Novelty/Model System for Author):

The manuscript presents the first patient identified carrying a mutation in the OXA1L gene, coding for a mitochondrial membrane protein insertase. In addition to samples from the patient (muscle, brain and fibroblasts), the authors have use additional models where they have implemented gene-editing approaches to study the function of OXA1L.

Referee #1 (Remarks for Author):

This manuscript describes the identification of pathogenic mutations in the gene OXA1L in patients suffering from mitochondrial encephalomyopathy. OXA1L codes for a mitochondrial inner membrane protein, member of the YidC/Alb3/Oxa1 family of protein insertases. Yeast Oxa1 has been shown to interact with the mitochondrial ribosome to facilitate orientation of the exit tunnel towards the membrane and co-translational insertion of newly-synthesized mitochondrial polypeptides. However, it is not clear whether the function of OXA1L is fully conserved from yeast to human, since previous studies have shown that knockdown of OXA1L in HEK293 cells affects the biogenesis of the F1F0-ATP synthase and complex I without altering the abundance of complex III or IV, whose mtDNA-encoded subunits are substrates of Oxa1 in yeast. However, the patient reported here presented with tissue-specific OXPHOS deficiencies. A deep isolated CIV deficiency in muscle and low levels of CIV subunits in muscle and fibroblasts, thus linking OXA1L function to CIV biogenesis. But also a CI deficiency in brain and fibroblasts. The authors extended their studies to two additional models, siOXA1 treated U2OS cells, and OXA1L-KO in HEK293T cells. Their analysis clearly showed a generalized effect on OXPHOS complexes and also on the abundance of mitochondrial ribosome markers. The effect on mitoribogenesis and mitochondrial translation was not observed in patient's fibroblast, probably due to the residual amount of functional OXA1L remaining.

The manuscript is technically and conceptually sound and I believe is up to the standards of EMBO Mol Med.

To complete the molecular characterization of the WT and variant OXA1L proteins, however, it would be important to include an additional experiment, where co-translational membrane insertion of mtDNA-encoded subunits (e.g. COX2 and some of the NDs, but all can be analyzed in the same experiment) would be monitored in the several models presented, in wild-type and OXA1L deficient cells. In a second experiment, the authors should assess the mitochondrial ribosome binding capacity of the WT and variant OXA1L proteins. With these data in hand, their discussion of human OXA1L function in health and disease will be enriched.

Finally, the tissue specificity of the OXA1L deficiency remains unexplained. The authors suggest the possibility of additional tissue-specific insertases with partially overlapping functions. However, this concept could be developed further, either by *in silico* searching for potential unknown insertases, or for example by testing if other known oxa1-family insertases (e.g. COX18) could play any overlapping role with OXA1L in a tissue specific manner.

Referee #2 (Comments on Novelty/Model System for Author):

The quality of several blots should be improved. There are some control missing. The characterization of the fly model is rather poor.

Referee #2 (Remarks for Author):

In this manuscript, Thompson and colleagues characterized the first patient with a mutation in OXA1L, a member of the insertase protein family, and investigated the effects of this mutation in tissues, cells and fly model.

Investigations on Oxa1p, the yeast orthologue of OXA1L, showed that it is a major player in the insertion of mitochondrially-encoded proteins into the inner mitochondrial membrane. In addition, it is involved in the biogenesis of the Tim18-Sdh3 module of the carrier translocase TIM22 complex, which is involved in the membrane insertion of mitochondrial carriers and of proteins imported via the TIM23-presequence translocase motor (PAM) and sorted to the inner mitochondrial membrane. Additional roles of Oxa1p include the assembly of respiratory complexes IV and V, the interaction with the mitochondrial ribosome. The complex findings by Thompson et al. suggest that the OXA1L

The description of the first patient is of great interest, but several points need to be clarified, as, in my opinion, not all the experiments are complete or conclusive.

By using WES, the authors described compound heterozygous mutations in OXA1L, leading to overall reduction of OXA1L protein expression. The patient presented with neurodevelopmental and metabolic impairment, degenerating into progressive encephalopathy and epilepsy leading to death by the age of 5. Histopathological and biochemical investigations of muscle biopsy revealed a complex IV deficiency, without major involvement of other complexes. Extensive

neuropathological characterization revealed several features, including microglial activation and gliosis, atrophy of the frontal lobe, cavitation of thalamus and striatum. Importantly, a prominent loss of complex I subunits was present in several areas of the brain, with the notable exception of the cerebellum.

Figure S2 and S3 are extremely dense and not very clear. No controls are shown and no markers are used to make the observed changes more evident. This makes the interpretation difficult for a reader without a specific background in neuropathology. The authors suggest a tissue-specificity in the observed effects on the respiratory chain, and I was wondering if they had the chance to measure the respiratory chain activities in the patient's brain, although I understand that it may not be easy. The tissue-specificity seems to be an important feature of the disease, as the fibroblasts displayed decreased COXI and II, NDUFB8, and complex V subunit ATP5B at steady state. As a general comment, the quality of the blots is in several cases rather poor and should be improved. There are also some inconsistencies: in figure 2A COXI and II are shown in fibroblasts, but only CII is shown for the muscle (Figure 2B). The assembly of complexes I, IV and V was affected in BNGE experiments in both skeletal muscle and cells, although only in cells a Complex I subassembly intermediate was observed. However, the characterization of the effects on respiratory chain complexes is quite superficial, and no in-depth analysis is shown, for instance no additional antibodies nor (eventually) 2-D BNGE are shown. In addition, no activities of the respiratory complexes in fibroblasts is shown.

Expression OXA1L via retroviral vector rescued complex IV and V proteins levels, but no data are shown on the complementation of the assembly or activities of respiratory chain complexes.

To substantiate their findings, the authors investigated a *Drosophila* model depleted of the fly orthologue of OXA1L. Reduced levels of Complex I proteins (NDUFS3) and oxygen consumption were observed. The blot however suggests a reduction also of complex II: is this the case? Again, the quality of the blots is rather poor, and does not allow a full interpretation of the results: as an example, tubulin and porin levels also seem higher in the mutant vs wild-type flies. Moreover, it is unclear whether whole flies were used for these experiments. Given the tissue-specificity observed in the patient, it would be of great interest to study if this is the case also in the fly, for instance by measuring OxPhos activities in the head vs the body.

siRNAs against OXA1L reduced the levels of COXII and mitochondrial ribosome proteins in U2OS cells. Only COXII antibody is shown, and it is difficult to infer a "clear complex IV defect" was present. Is this effect specific to complex IV or other complexes were also affected? Other antibodies and/or respiratory complexes activities/respiration should be investigated.

Disruption of OXA1L in HEK cells by CRISPR/Cas9 was not possible, likely because of the lethal phenotype of the mutant cells. However, the authors used a clever approach re-expressing a mutated form of the gene, not recognized by the gRNAs, under an inducible promoter. Induction of OXA1L expression prevented the lethal phenotype. Also in this case, depletion of OXA1L led to reduced levels of OXPHOS subunits and mitoribosomal proteins.

IP experiments were carried out to investigate the interactome of OXA1L. Several mtDNA-encoded proteins, including subunits of complex I, IV and V were found as well as a number of assembly factors.

Finally, the authors investigated mitochondrial translation in patient's fibroblasts, but did not observe changes in steady state levels of ribosomal proteins, while pulse and chase experiments revealed reduced stability of mitochondrial proteins, without changes in the synthesis. How do the authors reconcile this finding with the very low levels of mitoribosomal proteins? This sounds rather contradictory.

Referee #3 (Comments on Novelty/Model System for Author):

The study is very thorough and including retroviral complementation and knockout models (siRNA and CRISPR/Cas9, fly) and the results are ample and certainly sufficient to prove pathogenicity of novel mutations and confirms the importance of OXA1L for respiratory chain function and assembly.

Referee #3 (Remarks for Author):

This important work identifies the clinical biochemical and molecular characterization of a new form of mitochondrial disease defect due to biallelic variants of OXA1L, demonstrates and verifies

that OXA1 is crucial for the maintenance of mtDNA encoded respiratory chain subunits assembly of multiple respiratory chain complexes.

The study is very thorough and including retroviral complementation and knockout models (siRNA and CRISPR/Cas9, fly) and the results are ample and certainly sufficient to prove pathogenicity of the mutations and confirms the importance of OXA1L for respiratory chain function and assembly. Immunoprecipitation and experiments shows the association of OXA1L with respiratory chain complexes, assembly factors and shed light on its function. Translation experiments confirm the role of OXA1L in the translation and insertion process of mtDNA encoded proteins. Tissue specificity is also addressed.

I have only a few minor comments.

Case report: Was there any metabolic workup (other than lactate) performed, urine organic acids? Acylcarnitines? Amino acids?

Molecular genetics: Was any program used for the prediction of pathogenicity of the variants (mutation taster etc.), what were the scores?

OXPPOS steady state levels and assembly: How was loading performed according to protein or citrate synthase?

Translation; explain "residual OXA1L being higher than either knockdown experiment"

Discussion

Is there any explanation for normal CIII activity?

Re tissue specificity; Are there other protein with similar structure/sequence as that could be "insertase candidates" in the various tissues?

Materials and methods; add briefly, enzymatic assays and BNG methods

1st Revision - authors' response

22nd June 2018

EMM-2018-09060, Thompson *et al.*

Response to Reviewer's comments

Referee #1 (Comments on Novelty/Model System for Author):

The manuscript presents the first patient identified carrying a mutation in the OXA1L gene, coding for a mitochondrial membrane protein insertase. In addition to samples from the patient (muscle, brain and fibroblasts), the authors have used additional models where they have implemented gene-editing approaches to study the function of OXA1L.

Referee #1 (Remarks for Author):

This manuscript describes the identification of pathogenic mutations in the gene OXA1L in patients suffering from mitochondrial encephalomyopathy. OXA1L codes for a mitochondrial inner membrane protein, member of the YidC/Alb3/Oxa1 family of protein insertases. Yeast Oxa1 has been shown to interact with the mitochondrial ribosome to facilitate orientation of the exit tunnel towards the membrane and co-translational insertion of newly-synthesized mitochondrial polypeptides. However, it is not clear whether the function of OXA1L is fully conserved from yeast to human, since previous studies have shown that knockdown of OXA1L in HEK293 cells affects the biogenesis of the F1F0-ATP synthase and complex I without altering the abundance of complex III or IV, whose mtDNA-encoded subunits are substrates of Oxa1 in yeast. However, the patient reported here presented with tissue-specific OXPPOS deficiencies. A deep isolated CIV deficiency in muscle and low levels of CIV subunits in muscle and fibroblasts, thus linking OXA1L function to CIV biogenesis. But also a CI deficiency in brain and fibroblasts. The authors extended their studies to two additional models, siOXA1 treated U2OS cells, and OXA1L-KO in HEK293T cells. Their analysis clearly showed a generalized effect on OXPPOS complexes and also on the abundance of mitochondrial ribosome markers. The effect on mitoribogenesis and mitochondrial translation was

not observed in patient's fibroblast, probably due to the residual amount of functional OXA1L remaining.

The manuscript is technically and conceptually sound and I believe is up to the standards of EMBO Mol Med.

Author's Response: We would like to thank the reviewer for their thorough overview and kind comments regarding our manuscript.

To complete the molecular characterization of the WT and variant OXA1L proteins, however, it would be important to include an additional experiment, where co-translational membrane insertion of mtDNA-encoded subunits (e.g. COX2 and some of the NDs, but all can be analyzed in the same experiment) would be monitored in the several models presented, in wild-type and OXA1L deficient cells.

In a second experiment, the authors should assess the mitochondrial ribosome binding capacity of the WT and variant OXA1L proteins. With these data in hand, their discussion of human OXA1L function in health and disease will be enriched.

Author's Response: While we appreciate that there are many additional experiments that could be performed to further explore the molecular mechanisms of OXA1L in mitochondrial metabolism, we believe that the suggested experiments would be difficult to interpret. First, all mtDNA-encoded proteins are incredibly hydrophobic and would most likely aggregate if not inserted into the membrane (unless maintained by a chaperone(s)). The ³⁵S-met/cys *de novo* translation data from patient fibroblasts showed that after a 1hr pulse, there was no impairment of translation and that the newly synthesised mtDNA-encoded proteins were intact, strongly suggesting the proteins are highly likely to be inserted into the membrane in some way. The fact that after an 8hr chase the newly synthesised proteins in the patient fibroblasts have decayed more quickly than in control fibroblasts suggests to us that those proteins were either not inserted correctly (i.e. not in the correct orientation) or not assembled into their respective complexes correctly and were, therefore, likely to have been degraded. It is worth noting that we do not suggest all mtDNA-encoded proteins are directly inserted by OXA1L; nine of the 13 proteins are enriched in an OXA1L immunoprecipitation, but that does not necessarily indicate that all of these are direct interactions as there are significant interactions with many members of each complex. OXA1L is also likely to be a part of the assembly machinery.

The second experiment is also an interesting suggestion. However, our immunoprecipitation experiment with FLAG-tagged OXA1L does not pull down the whole mitoribosome, only a few specific subunits (Fig 4A). This implies that any primary interaction is likely to be weak. Further, the p.Cys207Phe amino acid substitution is not in the domain that reportedly binds the mitoribosome, which is the C-terminal tail (amino acids 334-436). Rather, the mutation is in a conserved transmembrane segment that in the YidC structure forms a hydrophilic groove, which is essential for substrate insertion across the lipid bilayer (Kumazaki *et al.* 2014: <https://www.ncbi.nlm.nih.gov/pubmed/24739968>, Dalbey and Kuhn 2014: <https://www.ncbi.nlm.nih.gov/pubmed/24799038>). Due to the high degree of structural conservation among this insertase family (Anghel *et al.* 2017: <https://www.ncbi.nlm.nih.gov/pubmed/29281821>), it is likely that the mechanism is relevant to OXA1L function. Therefore, the aromatic side chain of phenylalanine in this position may interfere with nascent chain interactions and thus impair protein function.

Finally, the tissue specificity of the OXA1L deficiency remains unexplained. The authors suggest the possibility of additional tissue-specific insertases with partially overlapping functions. However, this concept could be developed further, either by *in silico* searching for potential unknown insertases, or for example by testing if other known oxa1-family insertases (e.g. COX18) could play any overlapping role with OXA1L in a tissue specific manner.

Author's Response: We thank the reviewer for highlighting this (a similar comment was made by Referee #3) and we have sought to address this point. We have undertaken a detailed bioinformatic search for other potential insertases and for available expression data regarding *OXA1L*. These would suggest that only OXA1L and COX18 (previously known as OXA1L2) contain a YidC/Alb/Oxa1 domain. Recently COX18 was shown to be specifically needed to translocate the C-terminal tail of COXII across the inner membrane (<https://www.ncbi.nlm.nih.gov/pubmed/28330871>). There are no other known proteins with the

same domain other than OXA1L, COX18 and their respective isoforms, although recently more distant homologues have been found in the endoplasmic reticulum containing a DUF106 domain (Anghel et al 2017, <https://www.ncbi.nlm.nih.gov/pubmed/29281821>). These include WRB/Get1, TMCO1 and EMC3. We conducted a similar HHPred analysis that resulted in the same hits reported by Anghel et al. but with no other obvious candidates for related insertases (now included as **Appendix Table S2** and **Appendix Fig S1**). It would be interesting to see if any of these more distant Oxa1 homologues that are found in the endoplasmic reticulum have dual localisation, both in the ER and mitochondria. There are several different isoforms of *OXA1L* in humans and there is some evidence of differential tissue expression at the mRNA level (now shown in **Fig EV5** and **Appendix Table S1**), however there is no protein data available so we do not know whether these alternatively spliced isoforms produce a protein product. It is interesting to note that the brain appears to have the lowest relative *OXA1L* expression levels and is the only tissue we have tested in which we do not see an obvious complex IV defect. However, with the limited information available on isoform expression, particularly at the protein level, we cannot make any clear insights to explain the tissue specificity in this case. We have compiled the available bioinformatic data on the known *OXA1L* isoforms and homologues and added this to the manuscript as supplementary data (new **Fig EV5**, **Appendix Fig S1** and **Appendix Tables S1 and S2**).

Referee #2 (Comments on Novelty/Model System for Author):

The quality of several blots should be improved. There are some control missing. The characterization of the fly model is rather poor.

Author's Response: We regret that the reviewer feels some of the blots should be improved. We have tried to address this within a revised version and respond to some of the more specific points highlighted by the reviewer below.

Referee #2 (Remarks for Author):

In this manuscript, Thompson and colleagues characterized the first patient with a mutation in OXA1L, a member of the insertase protein family, and investigated the effects of this mutation in tissues, cells and fly model.

Investigations on Oxa1p, the yeast orthologue of OXA1L, showed that it is a major player in the insertion of mitochondrially-encoded proteins into the inner mitochondrial membrane. In addition, it is involved in the biogenesis of the Tim18-Sdh3 module of the carrier translocase TIM22 complex, which is involved in the membrane insertion of mitochondrial carriers and of proteins imported via the TIM23-presequence translocase motor (PAM) and sorted to the inner mitochondrial membrane. Additional roles of Oxa1p include the assembly of respiratory complexes IV and V, the interaction with the mitochondrial ribosome. The complex findings by Thompson et al. suggest that the OXA1L

The description of the first patient is of great interest, but several points need to be clarified, as, in my opinion, not all the experiments are complete or conclusive.

Author's Response: We thank the reviewer for their careful reading of our manuscript relating to OXA1L function. We are pleased that the reviewer feels the description of the first OXA1L patient is of great interest and aim to address each of the reviewer's concerns below.

By using WES, the authors described compound heterozygous mutations in OXA1L, leading to overall reduction of OXA1L protein expression. The patient presented with neurodevelopmental and metabolic impairment, degenerating into progressive encephalopathy and epilepsy leading to death by the age of 5. Histopathological and biochemical investigations of muscle biopsy revealed a complex IV deficiency, without major involvement of other complexes. Extensive neuropathological characterization revealed several features, including microglial activation and gliosis, atrophy of the frontal lobe, cavitation of thalamus and striatum. Importantly, a prominent loss of complex I subunits was present in several areas of the brain, with the notable exception of the cerebellum.

Figure S2 and S3 are extremely dense and not very clear. No controls are shown and no markers are used to make the observed changes more evident. This makes the interpretation difficult for a reader

without a specific background in neuropathology. The authors suggest a tissue-specificity in the observed effects on the respiratory chain, and I was wondering if they had the chance to measure the respiratory chain activities in the patient's brain, although I understand that it may not be easy.

Author's Response: We thank the reviewer for their comments; we have added some control staining as an inset to panels in **Fig EV2** and as separate panels (**Fi-iv**) in **Fig EV2** to make the observed differences more apparent. In relation to the specific respiratory chain activities, all of the brain tissue that was available to us was fixed (FFPE) and therefore measuring enzyme activities was not possible.

The tissue-specificity seems to be an important feature of the disease, as the fibroblasts displayed decreased COXI and II, NDUFB8, and complex V subunit ATP5B at steady state. As a general comment, the quality of the blots is in several cases rather poor and should be improved. There are also some inconsistencies: in figure 2A COXI and II are shown in fibroblasts, but only CII is shown for the muscle (Figure 2B).

Author's Response: We agree that tissue specificity is a well-recognised and important feature of mitochondrial disease in general, not just for the OXA1L patient described here. We regret that the reviewer considers some of the blots to be of low quality. We have replaced the muscle blot showing only OXA1L and COXII with a more complete panel including an antibody to a subunit of each OXPHOS complex (**Fig 2B**).

The assembly of complexes I, IV and V was affected in BNGE experiments in both skeletal muscle and cells, although only in cells a Complex I subassembly intermediate was observed. However, the characterization of the effects on respiratory chain complexes is quite superficial, and no in-depth analysis is shown, for instance no additional antibodies nor (eventually) 2-D BNGE are shown. In addition, no activities of the respiratory complexes in fibroblasts is shown.

Author's Response: It is true that we only saw a Complex I assembly intermediate in the fibroblast BNGE and not in the skeletal muscle. This matches the data we have from the respiratory chain activities in skeletal muscle where only CIV was markedly impaired, whereas CI activity was at the lower end of our established control range. We have now performed complex activity experiments in fibroblasts, which has also demonstrated that complex IV is the most affected of the complexes, although this activity is not as severely decreased as in skeletal muscle. Complex IV activity was increased by introducing a wild-type copy of OXA1L (now added as **Fig 2F**). We appreciate that we have not performed extensive BN PAGE analysis with alternative antibodies or 2D-BNGE, but feel that the main message is that there is a defect in the assembly of several OXPHOS complexes and that studying the composition of a subcomplex of CI is beyond the remit of this manuscript.

Expression OXA1L via retroviral vector rescued complex IV and V proteins levels, but no data are shown on the complementation of the assembly or activities of respiratory chain complexes.

Author's Response: We have shown that the steady state protein levels of complex IV subunits (COXI and COXII) and complex V subunit ATP5B are clearly lower in patient fibroblasts (**Fig 2A**) and skeletal muscle (**Fig 2B**) and that this correlates with a lack of fully assembled complex IV and complex V (**Fig 2C** and **2D**). Complex V activity is not routinely measured in our laboratory so we feel that the western blot analysis that shows both rescue of the steady-state levels of OXPHOS proteins (including complex V subunit ATP5B) as well as an increase in OXA1L itself is an acceptable way to demonstrate complementation of respiratory chain complex assembly. However, in response to the reviewer's comments and as mentioned above, we have now included data on the complex activities in fibroblasts, showing that complex IV activity is increased upon complementation with wild-type OXA1L (**Fig 2F**).

To substantiate their findings, the authors investigated a *Drosophila* model depleted of the fly orthologue of OXA1L. Reduced levels of Complex I proteins (NDUFS3) and oxygen consumption were observed. The blot however suggests a reduction also of complex II: is this the case? Again, the quality of the blots is rather poor, and does not allow a full interpretation of the results: as an example, tubulin and porin levels also seem higher in the mutant vs wild-type flies.

Author's Response: The levels of subunits of all OXPHOS complexes were decreased in the *Drosophila* model, as stated in the results '*Steady state levels of representative subunits from all OXPHOS complexes were decreased in OXA1L depleted flies compared to controls, with complex I (NDUFS3) being the most affected*'. Also the levels of tubulin were increased in the flies. This was consistent over many repeats and, crucially, porin levels were also increased, demonstrating that the loss of OXPHOS components was not purely due to lower mitochondrial mass. As we note in the discussion, we believe that the lower OXPHOS levels are potentially due to effects of OXA1L loss on the protein import machinery. Similarly, we believe that this is likely to explain why mitoribosomal proteins were decreased in the siRNA treated and CRISPR knockout cell lines ('*Problems with protein import would also account for the decreased SDHA seen in OXA1L depleted flies*' bottom page 13).

Moreover, it is unclear whether whole flies were used for these experiments. Given the tissue-specificity observed in the patient, it would be of great interest to study if this is the case also in the fly, for instance by measuring OxPhos activities in the head vs the body.

Author's Response: Whole flies were used for these experiments (we have now added this to the results for clarification, see page 9). We agree that the tissue specificity is of great interest in this case, but that this is also true for many mitochondrial genetic disorders. We do not feel that measuring OXPHOS activities in head vs body in these flies would enhance the study for several reasons; first, the *Drosophila* model does not model the specific mutation in the patient that demonstrates tissue specificity, but generically depletes the protein level following siRNA treatment. Second, as with the other severe depletions (siRNA or CRISPR/Cas9), of OXA1L in human cell lines there are secondary effects on MRPs and complex II that do not reflect the phenotype seen in the patient. It is, therefore, likely that a similar pattern would be observed in all tissues in the flies with all of the OXPHOS complexes being affected. Third, no further mechanistic insight would be derived even if the complex activities were to vary between the heads and bodies of the flies.

SiRNAs against OXA1L reduced the levels of COXII and mitochondrial ribosome proteins in U2OS cells. Only COXII antibody is shown, and it is difficult to infer a "clear complex IV defect" was present. Is this effect specific to complex IV or other complexes were also affected? Other antibodies and/or respiratory complexes activities/respiration should be investigated.

Author's Response: COXII was used as a marker for complex IV since depletion of COXII robustly causes assembly defects of the whole complex; for example, patients with *SCO1* or *SCO2* mutations – both of which encode proteins specifically required for COXII stability - demonstrate a loss of fully assembled complex IV/cytochrome *c* oxidase. We feel, therefore, that it is justifiable to infer a complex IV defect based on low COXII levels. The reason COXI was not in most of these analyses is that OXA1L and COXI migrate to a similar position on an SDS PAGE, so COXII was selected as a readout for complex IV status in most cases. To address an earlier point, we have now re-processed skeletal muscle samples and shown that both COXI and COXII levels are decreased to similar degrees in patient samples (**Fig 2B**), demonstrating a clear complex IV defect, confirming the use of COXII as a marker to infer complex IV integrity. For the siRNA depletion work in U2OS cells (**Fig 3C** and **3D**), the striking observation is that the mitochondrial ribosomal subunits are markedly affected, consistent with the OXA1L ablation work in HEK293 cells (**Fig 3G**). The decrease in MRPs will, of course, result in the depletion of mitochondrially encoded polypeptides, fully consistent with the loss of COXII, complex IV and other components of the respiratory chain. We believe the loss of MRPs is a secondary effect on major depletion of OXA1L as detailed in our discussion and highlighted again, above.

Disruption of OXA1L in HEK cells by CRISPR/Cas9 was not possible, likely because of the lethal phenotype of the mutant cells. However, the authors used a clever approach re-expressing a mutated form of the gene, not recognized by the gRNAs, under an inducible promoter. Induction of OXA1L expression prevented the lethal phenotype. Also in this case, depletion of OXA1L led to reduced levels of OXPHOS subunits and mitoribosomal proteins.

Author's Response: We thank the reviewer for their comments regarding the CRISPR/ Cas9 work.

IP experiments were carried out to investigate the interactome of OXA1L. Several mtDNA-encoded

proteins, including subunits of complex I, IV and V were found as well as a number of assembly factors.

Finally, the authors investigated mitochondrial translation in patient's fibroblasts, but did not observe changes in steady state levels of ribosomal proteins, while pulse and chase experiments revealed reduced stability of mitochondrial proteins, without changes in the synthesis. How do the authors reconcile this finding with the very low levels of mitoribosomal proteins? This sounds rather contradictory.

Author's Response: The reviewer correctly states that we did not observe any changes in mitoribosomal protein levels in patient fibroblasts. This is consistent with the synthesis of mt-DNA encoded proteins being unaffected in the patient.

The very low levels of mitoribosomal proteins was only observed in cells with very low OXA1L levels following CRISPR/ Cas9 knockout or siRNA treatment (see comments above and in discussion). We did not assess mitochondrial protein synthesis in these cells, but assume that it is severely impaired due to the lower MRP levels. These results are not contradictory, but instead reflect the difference in severity of the mutated OXA1L in the patient compared to very low/absent OXA1L in the CRISPR/Cas9 or siRNA treated cells.

Referee #3 (Comments on Novelty/Model System for Author):

The study is very thorough and including retroviral complementation and knockout models (siRNA and CRISPR/Cas9, fly) and the results are ample and certainly sufficient to prove pathogenicity of novel mutations and confirms the importance of OXA1L for respiratory chain function and assembly.

Referee #3 (Remarks for Author):

This important work identifies the clinical biochemical and molecular characterization of as new form of mitochondrial diseases defect due to biallelic variants of OXA1L, demonstrates and verifies that OXA1 is crucial for the maintenance of mtDNA encoded respiratory chain subunits assembly of multiple respiratory chain complexes.

The study is very thorough and including retroviral complementation and knockout models (siRNA and CRISPR/Cas9, fly) and the results are ample and certainly sufficient to prove pathogenicity of the mutations and confirms the importance of OXA1L for respiratory chain function and assembly. Immunoprecipitation and experiments shows the association of OXA1L with respiratory chain complexes, assembly factors and shed light on its function. Translation experiments confirm the role of OXA1L in the translation and insertion process of mtDNA encoded proteins. Tissue specificity is also addressed.

Author's Response: We thank the reviewer for their summary of our manuscript and the kind comments regarding the importance and thoroughness of our study.

I have only a few minor comments.

Case report: Was there any metabolic workup (other than lactate) performed, urine organic acids? Acylcarnitines? Amino acids?

Author's Response: Yes, several metabolites were measured in the patient as follows: lactate, pyruvate and the lactate/pyruvate ratio were increased; serum amino acids showed increased Alanine (520µmol/L; normal range < 416) in relation to the increased level of lactate. Ammonia, CDG and Biotinidase activity were normal, as was PDHc activity in patient fibroblasts. Acylcarnitines and urinary organic acids were not determined. These details are now included in a revised version of the manuscript text (page 5).

Molecular genetics: Was any program used for the prediction of pathogenicity of the variants (mutation taster etc.), what were the scores?

Author's Response: Due to the fact that one of the *OXA1L* variants causes a frameshift and the other affects splicing, ACMGG interpretation guidelines consider these loss of function alleles. However,

as we have demonstrated, the splice variant can also cause a missense change (Cys207Phe). We have shown that the Cys207 residue is moderately well conserved through evolution, but did not include pathogenicity predictions. The scores from prediction programs PolyPhen2 (score=0.055; benign) and SIFT (score=0.12; tolerated) which suggest that the Cys207Phe substitution would be tolerated, however these scores are based entirely on the predicted protein change and do not take into account the effect on splicing, which is what defines this variant as pathogenic. At the nucleotide level, MutationTaster predicts the variant to be disease causing and the variant has a CADD score of 24.9. The fact that there is approximately 30% residual OXA1L protein in patient fibroblasts confirms that the splicing defect leads to a functional consequence of decreased OXA1L protein even if the residual protein (with the Cys207Phe substitution) was fully functional. We have added some of this information to the manuscript (see page 6).

OXPHOS steady state levels and assembly: How was loading performed according to protein or citrate synthase?

Author's Response: Steady state protein levels were assessed by western blot using non-mitochondrial markers alpha tubulin or beta actin to normalise to protein levels. The assessment of assembly used complex II (SDHA antibody) as a loading control as complex II has no mtDNA component. The biochemical complex activity assays for skeletal muscle show the activity of each mitochondrial respiratory chain complex normalised to citrate synthase activity and the fibroblast complex activity data is shown normalised to complex II activity.

Translation; explain "residual OXA1L being higher than either knockdown experiment"

Response: This point refers to the patient having detectable levels of OXA1L on the western blot of approximately 30% of controls. This is different to the CRISPR and siRNA knockdowns of OXA1L which have very low or undetectable levels of OXA1L, hence in the patient the residual levels of OXA1L are greater than in either the CRISPR or siRNA experiments. We have changed the wording of this point to try to clarify this point further to avoid confusion. It now reads '*The stability of the mitoribosome in patient fibroblasts may be explained by the higher residual levels of OXA1L (approximately 30% relative to control fibroblasts) than in either of the knockdown experiments, where residual OXA1L levels are much lower.*' (see page 11).

Discussion: Is there any explanation for normal CIII activity?

Author's Response: CIII is, in our experience, commonly the least affected complex that contains a mtDNA encoded component; this is true in many cases of patients with defects in mitochondrial gene expression leading to a generalised problem with mt-translation – marked effects on complex I and IV are noted but not on complex III (i.e. see Thompson et al. *Am. J. Hum. Genet.* 2016; <https://www.ncbi.nlm.nih.gov/pubmed/27132592>). It is likely that OXA1L is not as important for the insertion or assembly of complex III subunits than it is for complex IV and V in particular.

Re tissue specificity; Are there other protein with similar structure/sequence as that could be "insertase candidates" in the various tissues?

Author's Response: As mentioned in our response to Referee #1, we have now included additional supplementary information detailing current knowledge on known OXA1L isoforms and homologues (new **Fig EV5**, **Appendix Fig S1** and **Appendix Tables S1 and S2**). Recently, more distant homologues of OXA1L have been shown to work in the endoplasmic reticulum. It is possible some of these proteins could have some localisation to mitochondria, or have tissue specific changes in expression.

Materials and methods; add briefly, enzymatic assays and BNG methods

Author's Response: Methods for the BNGE are already referenced in the methods section under '*Western blotting and Blue Native PAGE from patient tissues*' and '*Analysis of human OXA1L depletion*' as slightly different methods are used between the different centres contributing to this study. We have now added referenced methods for the complex activity enzymatic assays under '*Muscle Histology and Biochemistry*'.

Thank you for the submission of your revised manuscript to EMBO Molecular Medicine. We have now received the enclosed reports from the referees that were asked to re-assess it. As you will see the reviewers are now globally supportive and I am pleased to inform you that we will be able to accept your manuscript pending editorial final amendments.

REFeree REPORTS.

Referee #1 (Comments on Novelty/Model System for Author):

Several model organisms are used. Their analyses have provided compelling information regarding the pathogenic roles of OXA1L mutations.

Referee #1 (Remarks for Author):

The authors have responded to all my previous concerns.

Referee #2 (Remarks for Author):

The authors made an effort to improve the quality of the blots and to clarify the points I raised in my comments. I find this version very much improved. I wish only to point out that a better investigation of the fly model will not be irrelevant given the tissue specificity of the disease in humans. I am sure this will warrant future work and is beyond the scope of the present manuscript.

Referee #3 (Remarks for Author):

Revision satisfactory, no additional comments.

Corresponding Author Name: Robert W. Taylor

Manuscript Number: EMM-2018-09060